# Tracing near-surface runoff in a pre-Alpine headwater catchment

Anna Leuteritz[1*], Victor A. Gauthier[1*], Ilja van Meerveld[1]

[1]Department of Geography, University of Zurich, Zurich, Switzerland

* joined first authorship

*Correspondence to*: V.A. Gauthier (victor.gauthier@lilo.org) and A. Leuteritz (anna.leuteritz@geo.uzh.ch)

**Abstract.** Near-surface flow pathways (i.e., overland flow and topsoil interflow) play a crucial role in runoff generation and solute transport in steep humid catchments with low-permeability gleysols but remain understudied. We conducted sprinkling experiments on two large (>80 m$^2$) trenched runoff plots in the Swiss pre-Alps. One plot

was located in a natural clearing in an open mixed forest and the other in a grassland. After reaching steady state conditions, we applied uranine and NaCl to the surface as line tracers, injected NaBr into the subsurface (at ~20 cm depth), and added deuterium-enriched water via the sprinklers to assess the particle velocities for near-surface flow pathways and the interaction between overland flow and topsoil interflow. We compared these velocities with the celerity, which was determined by temporarily adding more water to the plots at different distances (2, 4 and

6 m) from the runoff collectors. To trace overland flow and determine its flow path lengths, we also applied brilliant blue dye at different locations on the surface of the plots.

The breakthrough curves highlight the rapid transport of water and solutes. The average (over all tracer applications) of the maximum velocities for overland flow and topsoil interflow were 51 m h$^{-1}$ and 30 m h$^{-1}$ for the plot in the clearing, and 24 m h$^{-1}$ and 17 m h$^{-1}$ for the plot in the grassland, respectively. The tracer breakthrough

curves highlight the interaction between overland flow and topsoil interflow as the NaBr that was injected in the subsurface mainly exited the plot in the clearing via overland flow. Similar to the results for the velocity, the celerity for overland flow was higher than for topsoil interflow at both locations. The celerity of overland flow was 2-3 times higher than the velocity for both locations. This was also the case for topsoil interflow in the grassland plot. For topsoil interflow in the clearing the celerity and velocity were relatively similar, which we

attribute to the importance of flow through large macropores. The overland flow pathways were relatively short for most locations (< 5 m) and confirmed the considerable interaction between overland flow and topsoil interflow as the dye often resurfaced a few meters below the initial infiltration points. Together, these results highlight the interaction between overland flow and topsoil interflow and the important role of macropores and soil pipes (particularly in forested areas) for the rapid transport of water and solutes from the steep, vegetated hillslopes to

the streams.

**1 Introduction**

Hillslope trench studies (e.g., Freer et al., 1997; Woods and Rowe, 1996), sprinkling experiments and tracer experiments (Buttle and McDonald, 2002; Meißl et al., 2021; Montgomery et al., 1997) have been used to investigate hillslope flow pathways in different environments. They have shown that subsurface flow can rapidly transport water and solutes downslope (e.g., Anderson et al., 1997; Feyen et al., 1999; Graham et al., 2010; Jackson et al., 2016; Tsuboyama et al., 1994; van Verseveld et al., 2017; Weiler et al., 1999; Wienhöfer et al., 2009) and that preferential flow pathways can deliver a considerable fraction of the total subsurface flow (e.g., Anderson et al., 2009a; Ehrhardt et al., 2022; Noguchi et al., 1999; Uchida et al., 2005; Vlček et al., 2017). Experiments with dye tracers have highlighted that even though individual macropores are short, they form long connected networks of preferential flow pathways (Noguchi et al., 1999; Sidle et al., 2000, 2001). In well drained hillslopes, subsurface flow often occurs at the soil-bedrock interface (e.g., Freer et al., 2002; Tani, 1997; Tromp-van Meerveld and McDonnell, 2006; Weiler et al., 2006), but in catchments with lower permeability soils, flow through the biomat (Sidle et al., 2007), the O-horizon (e.g., Brown et al., 1999) and more permeable topsoil (e.g., Schneider et al., 2014) may be the most important pathway for lateral flow.

In pre-Alpine and Alpine catchments, overland flow (OF) may be an important runoff generation mechanism during large or intense events (Meißl et al., 2023; Scherrer et al., 2007; Weiler et al., 1999). However, for undisturbed vegetated hillslopes it is generally not widespread. Instead, OF tends to infiltrate into the soil after flowing over the surface for a short distance, but only very few studies have actually studied the length of OF pathways in natural environments with tracers (Gerke et al., 2015; Maier et al., 2023) or based on temperature (Wolstenholme et al., 2020). Therefore, it is not clear how far OF travels over vegetated hillslopes and whether the water that infiltrates, mixes with the soil water or flows through preferential flow pathways further downslopes and then exfiltrates as return flow (RF; Dunne, 1978). Preferential flow pathways can be a major contributor to OF (Jones, 2010; Putty and Prasad, 2000). A few studies have shown that flow from preferential flow pathways consists mainly of water that was already stored in the soil (i.e., old water) (e.g., Bazemore et al., 1994; McDonnell, 1990) rather than precipitation (i.e., new water), but it can contain some precipitation as well (Bachmair and Weiler, 2012; Kienzler and Naef, 2008; Klaus et al., 2013).

To understand hillslope runoff processes, it is important to distinguish the propagation of hydrological signals from the movement of the water itself (McDonnell and Beven, 2014). Celerity refers to the speed at which a hydrological response (e.g., a pressure wave) propagates through the system and determines the timing of the runoff responses. The particle velocity describes the travel time of water and solutes through a system. The celerity is generally (much) higher than the particle velocity, since pressure waves can be transferred rapidly through the system (e.g., Torres et al., 1998), while individual water particles require considerably more time to move downslope (McDonnell and Beven, 2014). Both, celerity and velocity depend on hydrological connectivity, flow pathways, moisture conditions and rainfall inputs (Hallema et al., 2016; McGuire and McDonnell, 2010; Saco and Kumar, 2004). So far, there have been only a few combined studies on the celerity and velocity at the plot or hillslope scale (Rasmussen et al., 2000; Scaini et al., 2017; Torres et al., 1998; van Verseveld et al., 2017). These studies have shown that preferential flow pathways considerably influence the timing of surface flow responses and depend on vegetation (Bond et al., 2020; Monger et al., 2022). However, there is still a lack of field data about the celerity of near-surface flow pathways (Kienzler and Naef, 2008).

To better understand water and solute transport via near-surface flow pathways in humid catchments with low-permeability gleysols, we conducted sprinkling and tracer experiments on two trenched runoff plots (>80 m²) in the Swiss pre-Alps: one in a natural clearing in a mixed forest and the other in a grassland. Overland flow (OF), which includes biomat flow, and subsurface flow through the densely rooted topsoil (referred to as Topsoil Interflow, TIF) occur regularly (Gauthier et al., 2025). More specifically, the experiments were designed to:

1. Quantify the celerity and velocity of overland flow (OF) and topsoil interflow (TIF)

2. Determine the interaction between OF and TIF

3. Assess the typical length of OF pathways

## 2 Study site

### 2.1 Studibach catchment

The study was conducted in the Studibach catchment, a 20-ha pre-Alpine headwater catchment located in the Alptal in Switzerland (47.038° N, 8.723°E). The elevation ranges from 1,270 to 1,650 m above sea level. The climate is humid, with an average annual precipitation of about 2,300 mm $y^{-1}$. Precipitation is evenly distributed throughout the year (Stähli et al., 2021) and snowfall accounts for about 30% of the annual precipitation (Stähli and Gustafsson, 2006). About a quarter of the annual precipitation is delivered by precipitation with a 10-minute intensity exceeding 6 mm $h^{-1}$ (van Meerveld et al., 2018). The mean annual temperature is 5.7 °C (Stähli et al., 2021).

The topography is shaped by landslides and soil creep, with steep slopes (up to 69°) and flatter areas. The latter are wetter and dominated by grasslands and wetland vegetation. The drier and steeper parts are covered by open coniferous forests (Hagedorn et al., 2000; Figure 1). The upper part of the catchment is used as a pasture during the summer months.

The gleysols are underlain by flysch, a heterogeneous calcareous and sedimentary bedrock with a low permeability (Mohn et al., 2000). The gleysols have a high silt and clay content (>85%), a low permeability (Schleppi et al., 1998), and an average depth of about 1 m, ranging from 0.5 m on the steep hillslopes and ridges to 2.5 m in flatter areas (van Meerveld et al., 2018; Rinderer et al., 2014). Flow through the more permeable topsoil (25 to 40 cm deep) is much faster than flow through the clay due to the presence of macropores formed by roots and animal burrows (van Meerveld et al., 2018). The median (± standard deviation) of the saturated hydraulic conductivity of the surface, measured at eight locations in the lower part of the catchment using a 22 cm diameter double ring infiltrometer, was 76 ± 153 mm $h^{-1}$ (Wadman, 2023).

Groundwater tables are typically close to the surface (between 0 and 1.5 m deep; Rinderer et al., 2014) and the catchment responds quickly to rainfall events. Streamflow can increase by several orders of magnitude within minutes to hours (van Meerveld et al., 2018). Rinderer et al. (2016) found that for about half of the analyzed events, streamflow at the catchment outlet began to rise earlier than the groundwater. This type of response has been reported in other catchments (e.g., Beiter et al., 2020; Camporese et al., 2014; Gelmini et al., 2022; Pavlin et al., 2021) and is generally seen as an indication that precipitation falling on the channel or overland flow on near stream areas (rather than groundwater or subsurface flow) are responsible for the initial increase in streamflow during an event. However, a study from a neighboring catchment (Bujak-Ozga et al., 2024) revealed that the event water flux is much larger than the precipitation falling onto the stream network and therefore must come from

areas outside the flowing stream network, except at the beginning of the events. This suggests that fast surface or near surface flow pathways play a key factor in the catchment's runoff dynamics.

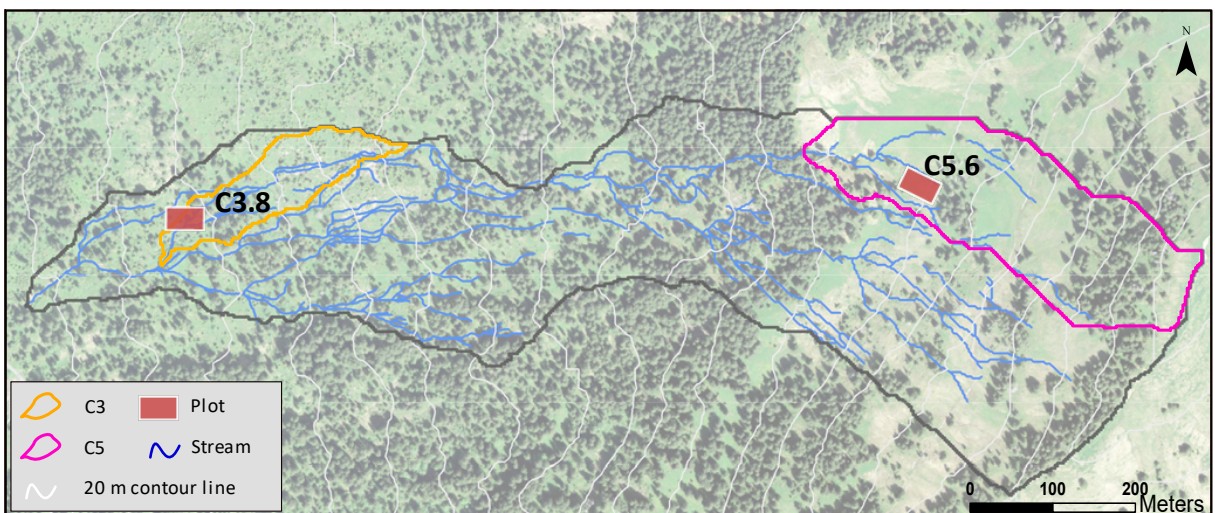

**Figure 1: Map of the Studibach catchment (black) with the location of subcatchment C3 (orange outline) and C5 (pink outline), and the runoff plots where the sprinkling and tracer experiments were done (red rectangles, not to scale). One plot (C3.8) is located in a natural clearing in the forest, while the other (C5.6) is located in a grassland that is used as a**

**pasture in summer. The grey lines represent the 20 m contour lines and the blue lines the mapped stream network. Background image: Swisstopo SwissImage (2023).**

### 2.2 Plot locations

Two locations with relatively straight slopes, at least 15 m wide and 15 m long were selected for the sprinkling

and tracer experiments (Figure 1; Table 1). One plot is located in a natural clearing in an open coniferous forest dominated by *Picea abies* (plot C3.8; Figure 2a) and the other (plot C5.6; Figure 2b) is located in a meadow that is used as a cattle pasture during the summer months. We refer to these plots as the clearing and grassland plot, respectively.

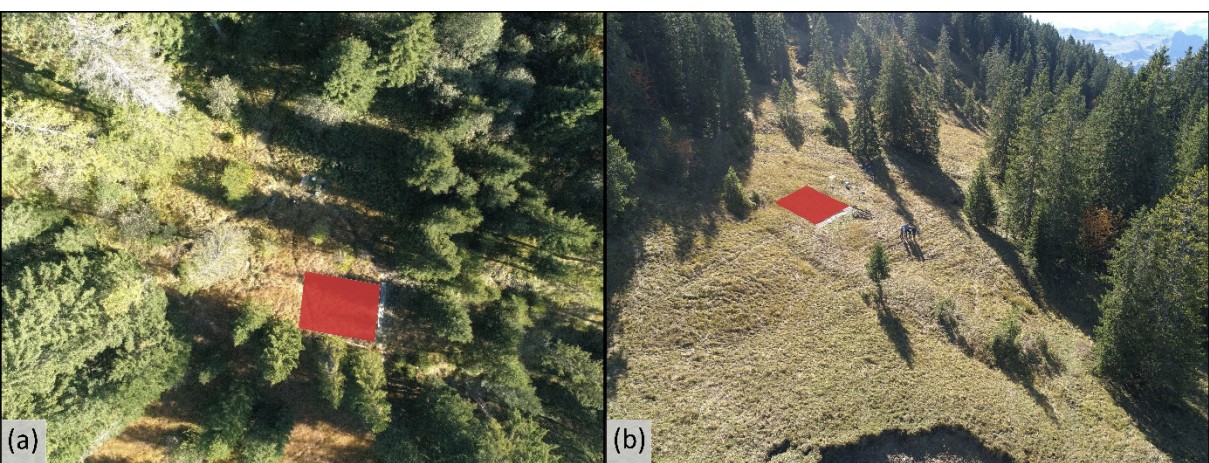

**Figure 2: Drone images of the plots in (a) the natural clearing in the open forest and (b) the grassland area. The location of the runoff plot areas is highlighted by the red trapezoids. For the location of the plots in the catchment see Figure 1.**

The vegetation on the plot in the natural clearing consists primarily of grasses, alpine flowers (*Chaerophyllum hirsutum*, *Lactuca virosa*, *Aconitum napellus*, and *Filipendula ulmaria*), and horsetails (*Equisetum* spp.). The mean slope is 9°. The surface is hummocky and contains some depressions. The soil profile consists of an A horizon underlain by a reduced Bg horizon. The A horizon consists of an upper 10 cm layer that is very rich in organic material, slightly decomposed and densely rooted, and a lower 30 cm topsoil layer, less rich in organic material (Table 1) that is more decomposed, and with fewer roots. The Bg horizon is composed of reduced clay with small stones (Ø <5 cm) and extends to at least 70 cm below the soil surface. The root density decreases with depth into the topsoil. The first 10 cm of soil contains many small roots and some big roots. There were many partially decomposed pieces of wood, such as old branches or small sections of trunk within the first 70 cm of soil. Most old roots were buried in the topsoil (up to 40 cm depth) and occasionally extended into the dense clay layer.

The vegetation of the grassland plot consists of horsetails (*Equisetum* spp.), small alpine flowers (*Succisa pratensis*, *Leontodon helveticus*, *Orchis mascula*), grasses, and scattered moss (*Selaginella helvetica*). The mean slope is 18°. The soil surface is more uniform than for the plot in the clearing. However, there were small "terraces and lobes", which could be attributed to solifluction and/or cattle trampling. The soil profile consists of an A horizon underlain by a reduced Bg horizon. The A horizon is an organic rich horizon up to 7 cm deep, and a topsoil horizon, composed of clay and organic material extending to 25 cm below the soil surface. The Bg horizon is composed of reduced clay that extended to at least 75 cm. Root density was highest in the upper 7 cm of the soil and decreased in the topsoil. The roots were small (Ø <0.5 mm). There were only a few large pieces of half-decomposed wood throughout the soil profile.

**Table 1: Overview of the plots and properties for the organic horizon (measured at 2-7 cm) and the topsoil (measured at 10-15 cm). The Topographic Wetness Index (TWI) is based on the calculations of (Rinderer et al., 2014) for a 6 m smoothed Digital Elevation Model. The slope was measured in the field. The porosity, moisture content at field capacity and drainable porosity are based on measurements for a soil core with the Hyprop (METER Group, USA). The organic matter content is based on the loss on ignition.**

|  | Clearing | | Grassland | |
|---|---|---|---|---|
| TWI | 7.0 | | 5.9 | |
| Slope | 9° | | 18° | |
| ***Soil depth*** | 2-7 cm | 10-15 cm | 2-7 cm | 10-15 cm |
| Soil bulk density (g cm$^{-1}$) | 0.21 | 0.23 | 0.53 | 0.38 |
| Porosity (%) | 85 | 84 | 80 | 79 |
| Moisture content at field capacity (pF 1.8) (%) | 68 | 65 | 70 | 73 |
| Drainable porosity | 17 | 19 | 11 | 6 |
| Organic Matter content (%) | 54 | 43 | 32 | 23 |

## 3 Methods

### 3.1 Runoff plots

#### 3.1.1 Plot setup

At the lower end of the ~80 m² plots, an eight-meter-long trench was excavated perpendicularly to the slope, following the methodology of Maier and van Meerveld (2021) and Weiler et al. (1999). Drain foil was placed along the trench face to block the lateral subsurface flow flowing through the topsoil. A drainage tube was wrapped in the foil and placed at the bottom of the 40 to 70 cm deep trench to collect the topsoil interflow (TIF) and route it to an Upwelling Bernoulli Tube (UBeTube). The trench was backfilled to ensure slope stability. An eight-meter-long gutter was installed on the surface and plastic foil was inserted into the soil (at ~3 cm depth on average) to guide the overland flow into the gutter. The water was then routed to another UBeTube via a hose. A fiberglass roof was installed over the gutter to prevent direct precipitation from entering the gutter.

The UBeTubes were built at the University of Zurich following the design of Stewart et al. (2015) using 10 cm diameter PVC pipe, in which a V-notch was cut with a water jet cutter (see Gauthier et al. 2025). A small piece of hose was attached to each UBeTube, just below the V-notch to facilitate the collection of water and to direct it to two boxes in which fluorescence sensors (Cyclops-7F Submersible Sensors, Turner Design, with a Cyclops-7 logger) were installed. A conductivity, temperature, and pressure logger (DCX-22-CTD, Keller Druck, Switzerland) was installed inside each UBeTube. To determine the water level from the pressure measurements, a barometric logger (DCX-22, Keller Druck, Switzerland) was placed outside the UBeTubes. The barometric logger was wrapped in a heat-reflecting foil to minimize temperature-related errors (Shannon et al., 2022). The loggers were set to a one-minute measurement interval. Laboratory-based rating curves were used to obtain the flow rate from the measured water levels.

At each runoff plot, we installed soil moisture sensors (TEROS 12 and GS3, METER Group) at 5, 15 and 25 cm below the soil surface at 2.5, 5, and 7.5 m from the trench. The sensors were connected to ZL6 and EM50 data loggers (METER Group) and recorded soil moisture at a 5-minute frequency.

#### 3.1.2 Sprinkler set up and rainfall measurements

For the rainfall simulation, we used Senninger I-Wob sprinklers (nozzle number 22) installed along the centre line of the plot at 2.5 m above the ground surface (Figure 3). These sprinklers are known to provide water with a relatively uniform spatial distribution and a representative raindrop size distribution (Maier and van Meerveld, 2021; van Meerveld et al., 2014). For the experiments in the clearing, stream water was applied to the sprinklers at 3 m and 7.5 m upslope from the trench. The stream water was collected from a location upstream and routed directly to the sprinklers via garden hoses (i.e., gravity driven; ~100 m elevation difference). For the experiments on the grassland plot, there was limited flow from the headwater streams and the water pressure was insufficient to run the sprinklers. Thus, for these experiments, we used a pump (MP2454, Dolmar, Germany). Because of this constraint, we only used one sprinkler, located 5 m from the trench (Figure 3).

At both plots, rainfall was recorded with two tipping bucket rain gauges (Davis Instruments Corp. with an Odyssey data logger; Dataflow Systems; 0.2 mm resolution) installed at 4.0 m and 6.5 m from the trench (Figure 3).

Additionally, we installed five rain collectors (funnel diameter: 100 mm) to determine the uniformity of the applied rainfall. The mean rainfall intensity was 24 mm h$^{-1}$ for the experiments in the clearing and 39 mm h$^{-1}$ for the experiments in the grassland (Table 2). These mean intensities correspond to intense rainfall events that occur on average one time per year for 24 mm h$^{-1}$, and every ten years for 39 mm h$^{-1}$ (maximum rainfall intensity recorded: 50 mm h$^{-1}$), based on 38 years of hourly precipitation data from the Erlenhöhe meteorological station, located 500 m from the Studibach outlet. As a daily rainfall amount of 100 mm occurs on average only every three years, the total amount of water applied over the experiments is extreme for the Alptal. Nevertheless, during natural rainfall, our sites become frequently near saturated and produce significant lateral water flow (Gauthier et al., 2025). Variations in mean intensity for the experiments in the clearing were attributed to small stones that partially obstructed the hose. For the experiments in the grassland, there were occasional issues with the pump or its power supply, leading to larger variations in the applied rainfall intensity.

## 3.2 Sprinkling experiments

### 3.2.1 Overview of the experiments

We conducted three different types of experiments on both plots: 1) water pulse experiments to determine the celerity, 2) tracer experiments to determine the velocity and mixing of OF and soil water, and 3) a blue-dye experiment to determine the length and shape of the OF pathways (Table 2). All experiments were conducted during steady state conditions, which were established by irrigating the plots until the OF and TIF rates were stable. There was a thin layer of snow (~5 cm) on the grassland plot prior to the first experiment. Some snow was carefully removed, and the plot was irrigated until no visible snow patches remained on the surface (and the OF and TIF rates were stable). Due to the limited number of daylight hours and nighttime temperatures falling below 0°C, overnight sprinkling was not possible for the grassland plot (Table 2).

**Table 2: Details of the different experiments for the plot in the natural clearing in the open forest and the plot in the grassland: date, sprinkling duration, and mean rainfall intensity ± standard deviation**

| Plot location and type of experiments | Date (dd.mm.yyy) | Duration of the experiment (h) | Mean intensity (mm h$^{-1}$) |
|---|---|---|---|
| *Clearing* | | | |
| Water pulse experiments | 08.08.2023 | 8.0 | 22 ± 2 |
| Tracer experiments | 09 and 10.08.2023 | 25 | 22 ± 2 |
| Blue dye experiment | 16.08.2023 | 4.0 | 28 ± 4 |
| *Grassland* | | | |
| Water pulse experiments | 09.11.2023 | 3.5 | 41 ± 5 |
| Tracer experiments | 08.11.2023 | 3.0 | 35 ± 13 |
| Blue-dye experiment | 09.11.2023 | 3.5 | 41 ± 5 |

### 3.2.2 Water pulse experiments

To determine the celerity of OF and TIF, we added additional water (~10 L min$^{-1}$) during the continued sprinkling after steady state flow conditions had been reached for both OF and TIF. The water was added across the plot at various distances from the trench (2, 4, and 6 meters) using an 8 m long hose with small holes (soaking hose) that

was suspended across the plot at ~50 cm above the surface. This additional "water pulse" increased the OF and TIF rates at the bottom of the plots above the steady state flow rates. Once a response was visually observed, the supply of the additional water was interrupted, and the system was allowed to return to the steady state flow rates before a new pulse was applied further upslope.

### 3.2.3 Tracer experiments

#### 3.2.3.1 Tracer application

After steady state was reached for both OF and TIF, NaCl and uranine were applied to the surface of the plots as line tracers, NaBr was added to the subsurface, and deuterium-labelled water was added via the sprinklers, (Table 3; Figure 3). All tracers were applied in solution. More specifically, in the clearing, we applied two lines of NaCl and uranine (named NaCl 1 and NaCl 2 and uranine 1 and uranine 2, respectively) by uniformly pouring the
230 dissolved tracer along a line across the plot within 1 minute (Table 3). The second line of NaCl and uranine tracers was applied 2.72 hours after the application of the first line, well after the peak concentrations had passed according to manual measurements of the Electrical Conductivity (EC) with a hand-held conductivity sensor (WTW Multi 3420, WTW Measurement Systems Inc). At the time of the first NaCl and uranine tracer application, we also applied 545 g NaBr to the subsurface via four 45 mm diameter PVC piezometers installed at 20 cm depth at 7.5 m
from the trench. For the plot in the grassland, we applied only one line of NaCl and uranine to the surface and 1500 g of NaBr to the subsurface via five piezometers installed at 6 m from the trench (Figure 3).

For the experiment in the clearing, we filled two ~500 L containers with stream water and added 150 mL of 70% deuterium water, yielding a $\delta^2H$ of 1516‰. The sprinklers were connected to these containers (and thus sprinkled deuterium-enriched (i.e., $D_2O$-labelled) water to the surface) for 30 min. For the grassland site, only one container
was filled with stream water. The addition of 150 mL of 70% deuterium water, yielded a $\delta^2H$ of 2604‰. The sprinklers applied the $D_2O$-labelled water to the surface of the grassland plot for 17 minutes.

**Table 3: Details of tracer experiments on the plots in the natural clearing in the open forest and the plot in the grassland.**

| Tracer experiment location and employed tracers | Type of application | Distance from trench (m) | Amount of applied tracer |
|---|---|---|---|
| *Clearing* | | | |
| NaCl 1 | Line application | 2.7 | 250 g in 4 L |
| NaCl 2 | Line application | 6.0 | 750 g in 4 L |
| Uranine 1 | Line application | 4.5 | 0.4 g in 4 L |
| Uranine 2 | Line application | 7.5 | 0.4 g in 4 L |
| NaBr | Subsurface injection | 7.5 | 545 g in 2.25 L |
| $D_2O$-enriched water | Surface application | - | 300 mL of 70% solution in about 1000 L |
| *Grassland* | | | |
| NaCl | Line application | 2.7 | 250 g in 4 L |
| Uranine | Line application | 4.5 | 0.4 g in 4 L |
| NaBr | Subsurface injection | 6.0 | 1500 g in 2.25 L |
| $D_2O$-enriched water | Surface application | - | 150 mL of 70% solution in about 500 L |

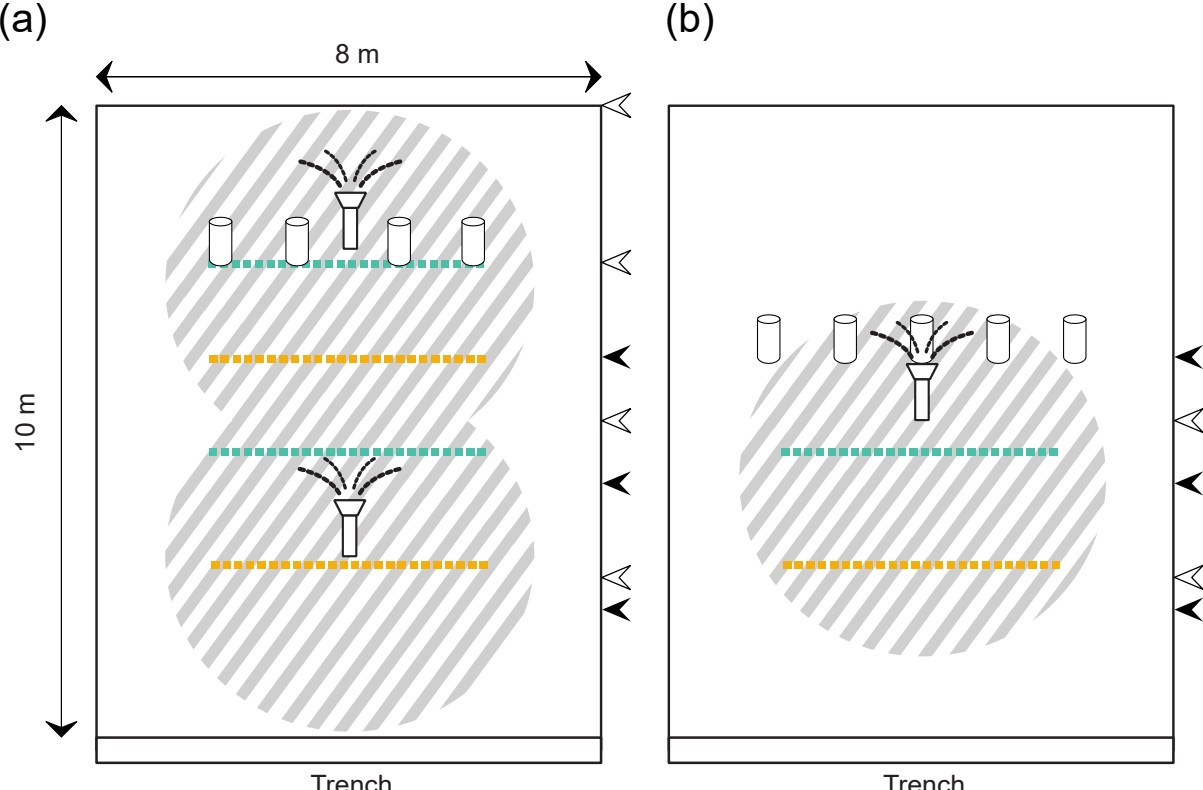

**Figure 3: Schematic overview of the tracer applications for the plots in the natural clearing (a) and in the grassland (b). Deuterium-labelled water (striped pattern) was applied to the surface via the sprinklers. NaCl (yellow) and uranine (green) were applied as lines at varying distances from the trench. NaBr was applied to the subsurface at a depth of 20 cm via piezometers. The location of the lines of the water pulses (black arrow heads) and blue dye (white arrow heads) are indicated on the side of the plots. The locations of the sprinklers are indicated with the sprinkler icons.**

### 3.2.3.2 Sample collection

During the tracer experiments, we manually collected samples of OF and TIF. For the experiment in the clearing, samples were collected at a one-minute interval for the first 41 minutes after the first tracer application (NaCl 1, uranine 1, NaBr, and $D_2O$-labelled water), followed by sampling every two minutes for 24 minutes, and sampling every five-minutes for 90 minutes. After the second line tracer application (NaCl 2 and uranine 2), samples were again collected at a one-minute interval for 40 minutes, followed by a five-minute interval for an additional 150 minutes. We used automatic samplers (model no. 6712, Teledyne ISCO, USA) to collect OF and TIF samples overnight at a one-hour interval. However, only the automatic sampler for OF functioned. Therefore, we manually collected three additional TIF samples the next day at intervals of one to two hours. In the grassland, the sampling intervals ranged between one and two minutes following the tracer application and continued for 2.5 hours. In addition to the sampling of OF and TIF, we sampled the stream that fed the sprinklers (every 1-2 hours for the experiments in the clearing and every 30 minutes for the experiments in the grassland). All samples were collected in 25 ml glass vials without headspace, stored in a fridge at 4°C, and filtered (0.45 µm SimplepureTM syringe filter) within a few days after sampling.

We recorded the electrical conductivity (EC) of OF and TIF with a Multi 3420 conductivity sensor (WTW Measurement Systems Inc) while we took the samples, and automatically (every minute) using the loggers (DCX-

22-CTD, Keller Druck, Switzerland) installed in the UBeTubes. The fluorescence sensor (see section 3.1.1) was calibrated prior to the experiments to obtain uranine concentrations at a 1-min interval.

### 3.2.3.3 Laboratory analyses

We analysed a selection of samples for bromide concentrations (see Table S1) at the Physics of Environmental Systems laboratory at ETH Zurich (Switzerland) using ion chromatograph (861 Advanced Compact IC, Metrohm AG). Another set of samples was analysed for the abundance of the stable isotopes of hydrogen and oxygen (from here on named stable water isotopes for brevity) using a cavity ring-down spectroscope (CRDS; L2140-i or L2130-i, Picarro, Inc.) at the Chair of Hydrology at the University of Freiburg, Germany. The analytical uncertainty is ± 0.6 ‰ for $\delta^2 H$.

### 3.2.3 Blue dye experiment

Brilliant blue dye was used to trace the OF pathways. As for the other experiments, rainfall was applied to the plots until steady state conditions were reached for both OF and TIF. The blue dye solution was manually applied along the surface of the plot as a line at 2.5, 5, 7.5, and 10 m upslope from the trench in the clearing, and at 2.5 and 5 m from the trench in the grassland (Figure 3). For each line (~ 6 to 7-m-long), we used two 1.5 L bottles with a concentration of 3 mg $L^{-1}$ of brilliant blue dye and applied it as uniformly as possible within a minute. Immediately following the dye application, the OF pathways were marked using three types of flags: one indicating the flowpath from the application to where it infiltrated into the soil, another indicating exfiltration (i.e., reappearance) of the dye, and a last one marking the second point of re-infiltration. Once the flow pathways had all been marked, the pattern of the OF pathways was sketched in a notebook using a 25 cm grid and photographs were taken with a drone (Phantom II, DJI) to complement the manual sketches.

In the clearing, the tall grass was cut 7 days prior to the blue dye experiment (but after the water pulse and tracer experiments) to be able to observe the flow pathways. In the grassland, this was not necessary because the experiment took place in November when the vegetation was not so tall, allowing us to see the soil surface.

### 3.3 Data analyses

### 3.3.1 Hydrometric responses

Flow rates were calculated from the water levels in the UBeTubes based on rating curves developed in laboratory ($Q = \alpha h^\beta$, where $\alpha = 0.24 \pm 0.08$, $\beta = 1.88 \pm 0.27$, $h$ is the water level above the bottom of the V-notch (in cm), and Q the flow rate (in L $min^{-1}$)). The estimated uncertainty in the flow rate is 12% at low flow rates (< 2 L $min^{-1}$) and 5% at higher flow rates. Runoff ratios were calculated by dividing the total amount of OF or TIF by the total rainfall. All analyses for the flow data were done in Python (version 3.12), using the packages *Pandas*, *Scipy*, *Matplotlib* and *Seaborn*.

### 3.3.2 Arrival times

The celerity and particle velocity were based on the time of the first increase in the water level in the UBeTube after the application of the water pulse and the first arrival of the tracer (i.e., timing of the sample with a concentration above the background concentration) respectively. They are thus the maximum celerity and velocity. There were some fluctuations in the water level due to changes in the sprinkling rate (Table 2 and Figure 5), so we took the sprinkling rates into account to find the first increase in water level and flow due to the application of the water pulses. No adjustments were made for potential delays caused by the transfer of water through the drainage

system or the gutter as these were assumed to be small. However, to determine the uncertainty of the celerity and velocity, we assumed an uncertainty in the timing of 2 minutes and uncertainty in the distance of 0.1 m.

### 3.3.3 Tracer recovery

One-minute time series of the bromide concentrations and $\delta^2H$ were generated by linear interpolation between the measurements. The background concentrations were subtracted from the measured concentrations to obtain the breakthrough curves. The background electrical conductivity (EC) and uranine concentrations for OF and TIF were based on the average of the measurements after steady state conditions were reached and 15 minutes before the first salt and uranine applications. Background concentrations of bromide were below the detection limit (0.001 mg L$^{-1}$). A laboratory calibration was used to convert the EC minus the background EC to NaCl concentrations.

The recovered tracer mass was estimated by integrating the mass fluxes (concentration minus the background concentration multiplied by the flow rate). For the calculations of the tracer recovery, we assumed that the sprinklers did not apply any uranine or bromide as these concentrations were all below the detection limit. We, similarly, assume that the chemistry of the irrigation water had a negligible effect on the chemistry of OF and TIF, given that the plots were irrigated for several hours a day before the experiments and the EC in OF and TIF remained relatively stable during this time, even though the stream water that was used for sprinkling had a higher EC than OF and TIF after steady state conditions were established. We expected that the NaBr that was added to the subsurface would take considerably more time to reach the OF and TIF collection systems than the NaCl that was added to the surface closer to trench (see Figure 3), and that interference with the EC measurements would be minimal. However, the breakthrough of NaBr was quick as well (see results section 4.3). Still, the peak concentration of NaBr would have increased the EC by only 2 µS cm$^{-1}$. This is comparable to the 1 µS cm$^{-1}$ resolution of the sensor and leads to an overestimation of the calculated NaCl concentrations by about 1 mg L$^{-1}$. We considered this overestimation acceptable considering all other uncertainties and thus did not correct the EC based estimates of the NaCl concentrations for the NaBr concentrations. Still collectively, these assumptions lead to some uncertainty in the recovery of the NaCl tracer.

We report the tracer recovery rates for the first 100 minutes to allow for a comparison between the two plots, as sampling for the grassland plot was limited to 100 minutes after tracer application. For the clearing, we additionally report the recovery until the time of the second line tracer application (NaCl 2 and uranine 2) and the end of the experiment (24 hours). The estimated uncertainty in the flow rates (see section 3.3.1) leads to a considerably larger uncertainty in the recovered mass than the uncertainty in measured tracer concentration. Thus, we did not consider the uncertainty in the concentrations in the uncertainty of the tracer recovery.

### 3.3.4 Two-component mixing model

We applied a two-component mixing model to estimate the fraction of the deuterium-labelled water ($f_e$) in OF and TIF:

$$f_e = \frac{(C_S - C_{pe})}{(C_e - C_{pe})}$$

where $C_S$ is the $\delta^2H$ for the OF or TIF sample, $C_{pe}$ is the $\delta^2H$ of the OF and TIF prior to the application of the labelled water (mean of -65.45 ‰ and -64.93 ‰ for OF and TIF for the experiment in the clearing and -77.16 ‰ and -78.8 ‰ for OF and TIF for the experiment in the grassland, respectively), and $C_e$ is the $\delta^2H$ for the labelled

sprinkler water (1516 ‰ and 2604 ‰ for the experiments in the clearing and grassland, respectively). Note that due to the considerable amount of water applied to the plots to test the sprinklers, reach steady state flow conditions

(and for the experiment in the clearing also for the water pulse experiments), the $\delta^2H$ of the OF or TIF samples collected right before the application of the labelled water were similar to those of the applied unlabelled water (-67.7‰ for the plot in the clearing and -73.1‰ for the grassland plot).

## 4. Results

### 4.1 Response of the plots to the sprinkling

In the clearing, steady state conditions were reached after on average 35 minutes of sprinkling for OF and 51 minutes for TIF. Soil moisture increased on average (for the three locations in the plot) by 10% at 5 cm, 1% at 15 cm and 5% at 25 cm during this time (Table S2). The flow rate during the steady state conditions was almost twice as high for TIF than OF (Figure 4a), with runoff ratios during the steady state conditions of ~ 20% for OF and ~ 46% for TIF. The remaining 34% of the water either percolated deeper into the soil or left the plot laterally as it

was not bounded. During the water pulse experiments, the OF and TIF flow rates were relatively constant, except during and following the application of the water pulses (Figure 5). During the tracer experiment, the flow rate was constant for OF, but for TIF there was a 25% increase in flow between 375 and 500 minutes. As there was no change in the sprinkling intensity, we think that this increase is mainly caused by changes in the boundary conditions, particularly on the right side of the plot (looking upslope) where a long surface flowpath was observed

during the blue dye experiments (see section 4.4). As we walked along this side of the plot and the soil on this side of the plot became very muddy, we may have influenced this flow pathway (e.g., temporarily blocked part of it).

For the grassland plot, steady state conditions were reached after on average 30 minutes of sprinkling for OF and 26 minutes for TIF. Soil moisture content increased minimally during this time (Table S2). The OF rate was much larger than for TIF, with runoff ratios of 44% and 5%, respectively (Figure 4b). Because of the use of the pump

and occasional issues with the power supply (e.g., to refill the petrol), the rainfall rate (Table 2) and flow rates fluctuated more than for the experiments in the clearing. There was a decline in the precipitation intensity and flow rates at 90 and 140 minutes after switching the source of the sprinklers for the tracer experiments (Figure 4b) and at 20 and 135 minutes during the water pulse experiment (Figure 5).

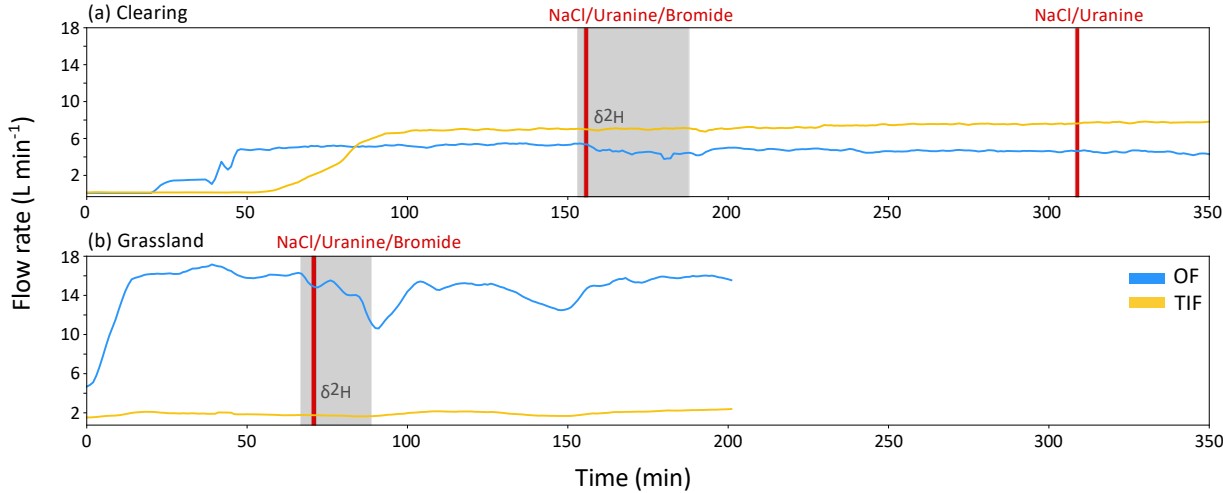

**Figure 4: Time series of the OF (blue) and TIF (orange) flow rates (L min⁻¹) during the tracer experiments on the plot in the clearing (a) and the plot in the grassland (b). The period when deuterium enriched water was applied is indicated by the grey shading. The times of the NaCl, uranine, and NaBr (bromide) tracer applications are indicated by the vertical red lines.**

### 4.2 Water pulse experiments

In the clearing, the water pulses at 2 m and 4 m from the trench produced a clear response for OF and TIF. The response was less pronounced for the pulse applied at 6 m from the trench (Figure 5a). Water was also applied at 8 m, but did not lead to a measurable response (data not shown). The increase in the flow above the steady state flow rate was more than four times larger for TIF than OF (Figure 5a). The calculated celerities were higher and more variable (depending on the distance from the trench) for OF (mean ± standard deviation: $150 \pm 80$ m h⁻¹) than for TIF ($34 \pm 5$ m h⁻¹), in part due to the very high celerity (240 m h⁻¹) from the water pulse applied at 4 m from the trench (Table 4). The celerity for OF was on average of a factor 4 times (range: 2.8-8.0) higher than for TIF. Note that the use of the ± 2 minutes uncertainty for the first response (2 measurements) was similar to the response time, leading to the very high uncertainties for the celerity of OF (Table 4).

In the grassland, all three water pulses (at 2 m, 4 m and 6 m) produced a clear flow response. In contrast to the results for the plot in the clearing, the increase in OF was much larger than for TIF (Figure 5b). The calculated celerities for OF (mean ± standard deviation: $64 \pm 7$ m h⁻¹) and TIF ($41 \pm 10$ m h⁻¹) were more similar for the three application distances than for the experiments in the clearing (Table 5). The celerity for OF was on average almost a factor two (range: 1.3-2.0) higher than for TIF. The celerity for OF in the grassland was, however, almost two times smaller for the grassland than for the clearing (when excluding the high celerity for the experiment at 4 m in the clearing). The celerities for TIF were more similar, though somewhat higher in the grassland than the clearing (Table 4).

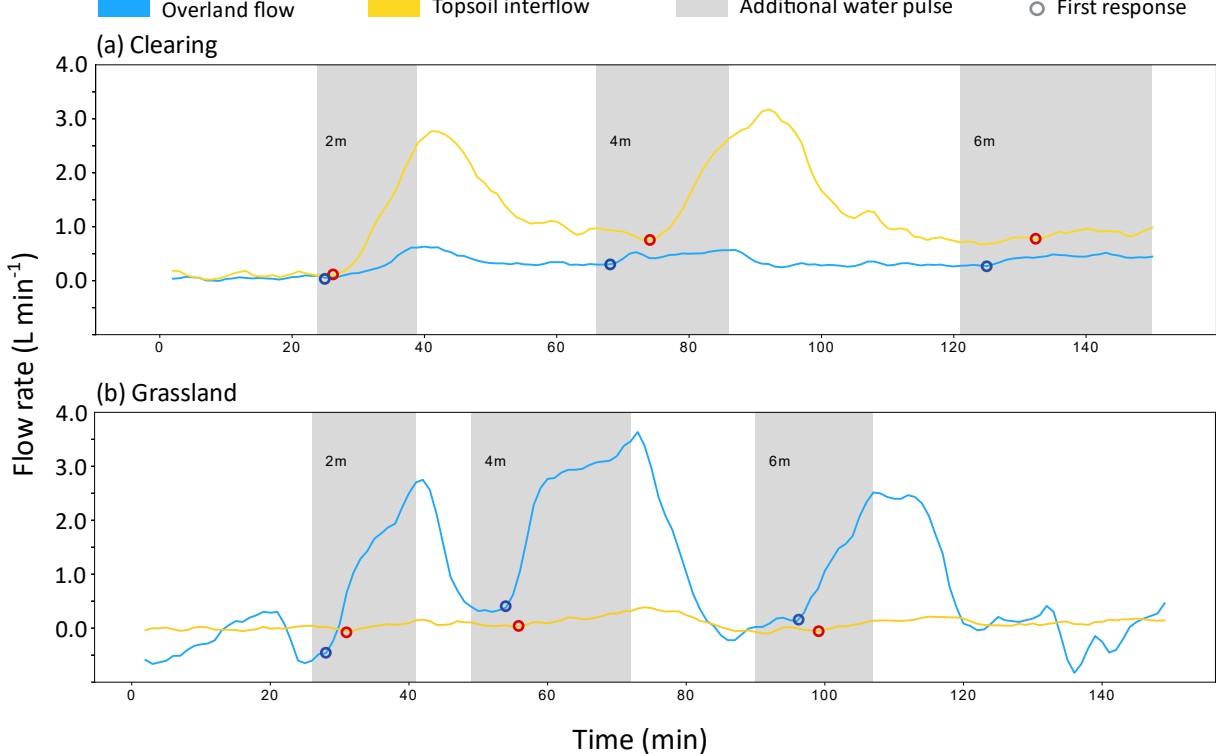

**Figure 5: Time series for the increase in flow rates above the average steady state flow rate for OF (blue) and TIF (orange) during the water pulse experiments for the plot in the clearing (a) and the plot in the grassland (b). The times during which the additional water pulses were added are indicated with the grey shading. The time of the first increase in the flow rate in response to the water pulse (used for the calculation of the celerity) is indicated with a circle (blue for OF and red for TIF).**

**Table 4: Calculated celerity ± the estimated uncertainty for the water pulses applied at different distances from the trench for overland flow (OF) and topsoil interflow (TIF) for the plots in the clearing and the grassland, as well as the average value ± standard deviation for each plot. The uncertainty is based on an uncertainty of 2 minutes for the timing and an uncertainty of 0.1 m for the distance.**

| Distance from trench (m) | Celerity ± uncertainty (m h⁻¹) | |
|---|---|---|
| | **OF** | **TIF** |
| *Clearing* | | |
| 2 | 120 ± 240 | 40 ± 10 |
| 4 | 240 ±480 | 30 ± 11 |
| 6 | 90 ±45 | 33 ± 9 |
| Average ± st. dev | 150 ± 80 | 34 ± 5 |
| *Grassland* | | |
| 2 | 60 ± 60 | 30 ± 7 |
| 4 | 60 ±19 | 48 ±8 |
| 6 | 72 ±20 | 41 ± 11 |
| Average ± st. dev | 64 ± 7 | 41 ±10 |

### 4.3. Tracer experiments

#### 4.3.1. Breakthrough curves and particle velocities

The line tracers (NaCl and uranine) appeared within 3-13 minutes after application, depending on the distance from the trench that they were applied (Figure 6). The deuterium-labelled water also appeared quickly in OF and TIF. It peaked after 36 minutes for OF and 112 minutes for TIF for the plot in the clearing and after 18 minutes for OF and 50 minutes for TIF for the plot in the grassland (Figure 6). The NaBr that was applied to the subsurface mainly appeared in OF for the plot in the clearing, arriving after 9 minutes for OF and after 27 minutes for TIF. For the plot in the grassland, NaBr concentrations remained below the detection limit.

The calculated maximum particle velocities were generally higher for OF than TIF and higher for the plot in the clearing than the plot in the grassland (Table 5). The average (± standard deviation) of the maximum particle velocities (calculated for the different tracers) for the plot in the clearing was $51 \pm 14$ m h$^{-1}$ for OF and $30 \pm 9$ m h$^{-1}$ for TIF. For the experiments on the grassland plot, the average of the maximum particle velocities was $24 \pm 1$ m h$^{-1}$ for OF and $17 \pm 6$ m h$^{-1}$ for TIF (Table 5). Thus, the velocities were higher for the plot in the clearing than the plot in the grassland.

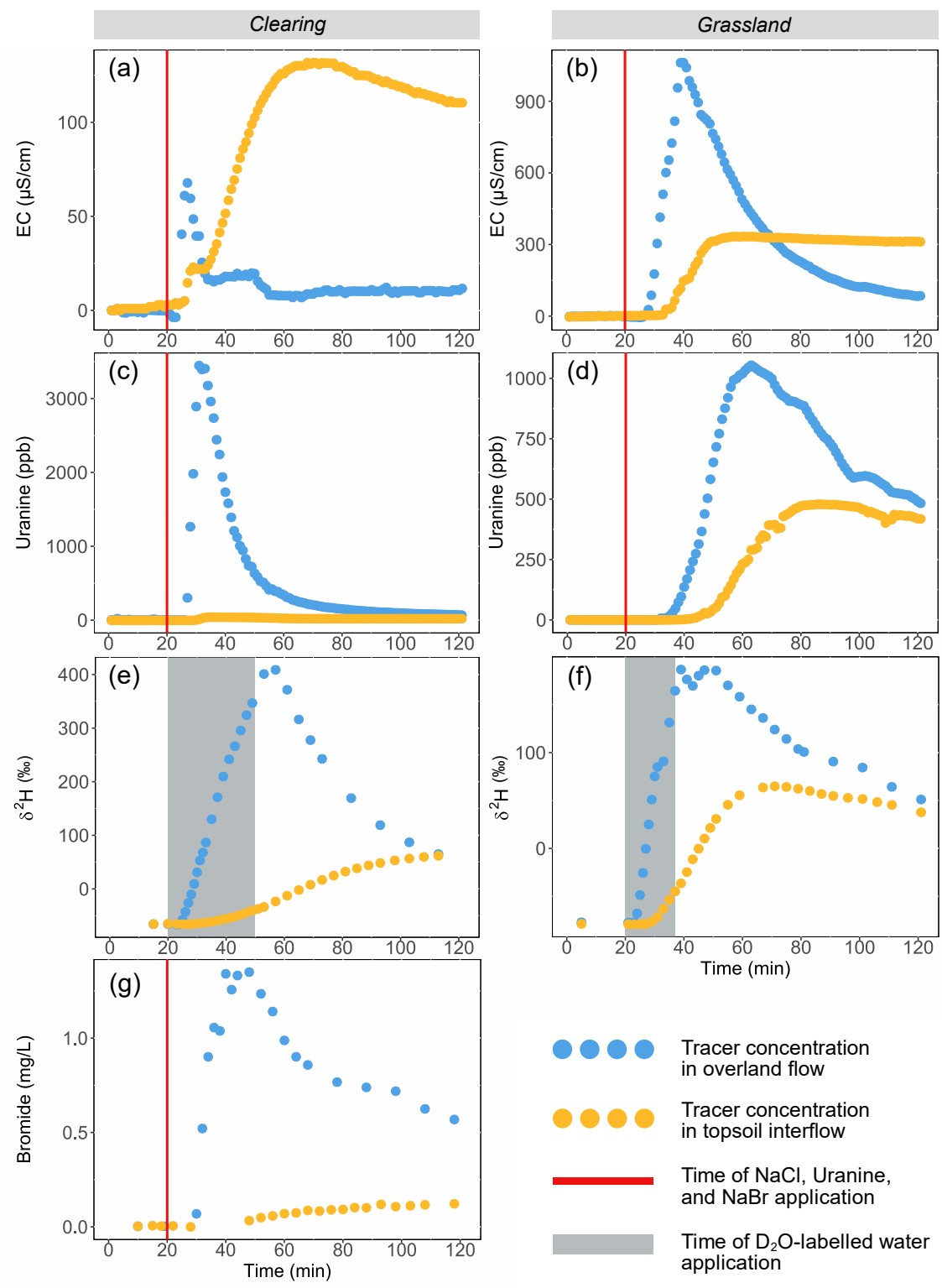

**Figure 6: Breakthrough curves for the first lines of NaCl (EC minus background EC; first row) and uranine (second row), the deuterium labelled water added via the sprinklers (δ²H; third row), and the NaBr applied to the subsurface (fourth row) for OF (blue) and TIF (orange) for the plot in the natural clearing (left column) and the plot in the grassland (right column). The bromide concentrations for the grassland plot remained below detection limit and are therefore not shown. Note that time 0 in panel a, c, e, and g correspond to minute 138 in Figure 4a, while time 0 in panels b, d, f corresponds to minute 50 in Figure 4b. The vertical red lines indicate the time of NaCl, uranine, and NaBr**

**application. The grey shaded areas indicate the time of the deuterium-labelled water application. For the responses for**
**the second line tracer applications, see Figure S1.**

**Table 5: Maximum particle velocities ± the estimated uncertainties for the tracers applied at different distances from the trench for overland flow (OF) and topsoil interflow (TIF) for the plots in the clearing and the grassland, as well as the average ± standard deviation for each plot. The uncertainty is based on an uncertainty of 2 minutes for the timing**
**and an uncertainty of 0.1 m for the distance. BDL stands for "below detection limit".**

| Location and tracer | Application (m from trench) | Maximum particle velocity ± uncertainty (m h$^{-1}$) | |
|---|---|---|---|
| | | OF | TIF |
| *Clearing* | | | |
| NaCl 1 | 2.7 | 54 ± 38 | 27 ± 10 |
| NaCl 2 | 6.0 | 72 ± 30 | 40 ± 10 |
| Uranine 1 | 4.5 | 45 ± 16 | 30 ± 7 |
| Uranine 2 | 7.5 | 35 ± 6 | 35 ± 6 |
| NaBr | 7.5 | 50 ± 12 | 17 ± 2 |
| Average ± st. dev | | 51 ± 14 | 30 ± 9 |
| *Grassland* | | | |
| NaCl | 2.7 | 23 ± 8 | 13 ± 2 |
| Uranine | 4.5 | 25 ± 5 | 21 ± 4 |
| NaBr | 6.0 | BDL | BDL |
| Average ± st. dev | | 24 ± 1 | 17 ± 6 |

### 4.3.2. Two-component mixing model

For the experiment in the clearing, the deuterium-labelled water appeared after only 3 minutes in OF and after 11 minutes in TIF. A larger portion of the labelled water left the plot as OF than TIF, despite the flow rate being 35%
lower for OF than TIF (Figure 7). The maximum fraction of labelled water in OF was 30% at 36 minutes after the start of the application. The average fraction of labelled water in OF during the first 100 minutes of the experiment (including the 30-minute application period) was 15%. In contrast, the fraction of labelled water in TIF increased gradually, reaching a maximum of 8% at 112 minutes after the start of application o (i.e., 82 minutes after the end of the application).

In the grassland, the labelled water appeared in OF after 2 minutes and peaked at 10% at 19 minutes after the start of the application. Similar to the experiment in the clearing, the labelled water mainly left the plot as OF. The average contribution of the labelled water to OF during the first 100 minutes, however, was lower than for the clearing (7% vs 15%). The contribution of the labelled water to TIF was small, with a maximum of 5% at 51 minutes after the start of the application (i.e., 34 minutes after the end of the application) and an average of 4% for
the first 100 minutes of the experiment (Figure 7). The response was, however, fast with the first arrival of the labelled water after just 6 minutes.

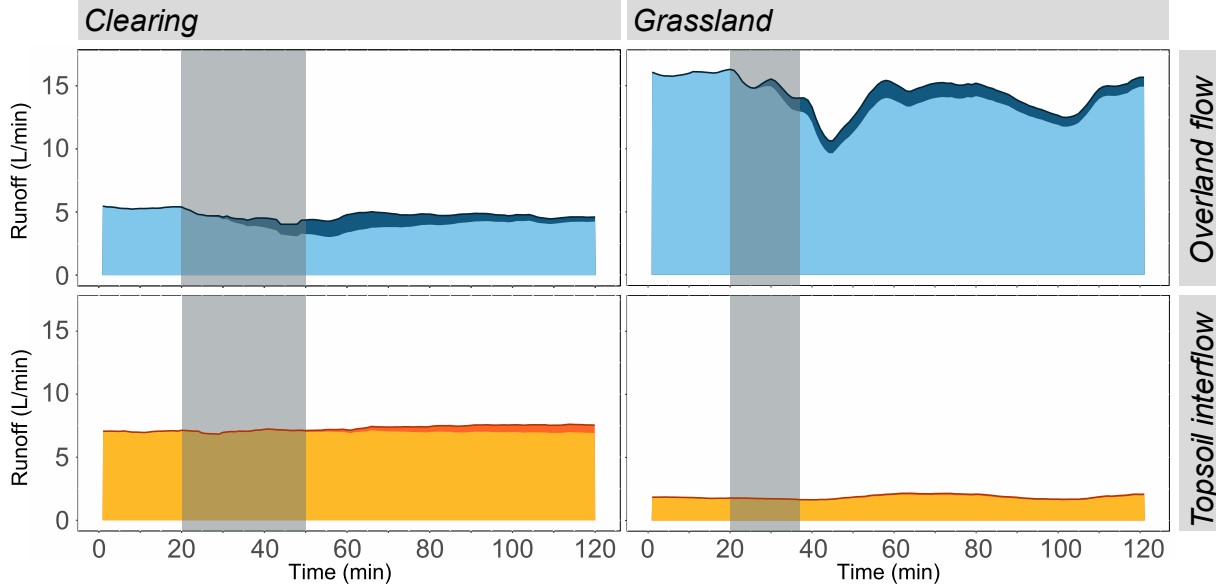

**Figure 7: Overland flow (top row, blue) and topsoil interflow (bottom panel, orange) rates, with the contribution of the deuterium-labelled water represented by the darker shades of blue and orange, respectively for the plot in the clearing (left) and the plot in the grassland (right). The grey shading indicates the time of the application of the deuterium-labelled water. Note that time 0 in the figure for the clearing (left) corresponds to minute 138th minute in Figure 4a, and for the plot in the grassland (right) to minute 50 in Figure 4b.**

### 4.3.3. Tracer recovery

For the plot in the clearing, most of the applied NaCl left the plot as TIF (Table 6; Figure 4), whereas most of the uranine that was applied only a few meters further upslope (Figure 3), left the plot as OF. The recovery of the NaBr, which was applied to the topsoil, was minimal after 100 minutes (<1% for both flow pathways). The recovery of uranine and NaCl was much higher for the plot in the grassland than for the plot in the clearing (107% for the grassland vs 14% for the clearing for NaCl, and 104% vs 27% for uranine) and more similar for the two flow pathways (Table 6), despite the much higher flow rate for OF than TIF. The tracer recoveries exceeding 100% in the grassland are largely attributed to the uncertainties in the flow measurements. The recovery of the deuterium-labelled water after 100 minutes was more similar for the two plots but still twice as high for the plot in the grassland than the plot in the clearing: 24% for the grassland vs 12% for the clearing. After 24 hours, 39% of the labelled water was recovered for the plot in the clearing, with about one-third of the labelled water leaving the plot as OF and two-thirds as TIF (Table 6).

**Table 6: Cumulative tracer recovery as percentage of the applied mass for each tracer used in the experiments for the plot in the clearing and the plot in the grassland. For the plot in the clearing, the second lines of NaCl and uranine were applied 163 minutes after the first applications. Therefore, the values reported for 100 and 163 minutes include only the recovery of the first tracer application, while the values reported for 24 hours include the recovery from both applications. Because the total applied mass increased with the second application and the recovery for the second line (applied further upslope) was less than for the first line, the cumulative recovery after 24 hours expressed as a percentage of the total applied mass is less than after 163 minutes. Some of the tracer applied to the upper parts of the plot in the clearing likely left via an outflow on the side of the plot (see section 4.4). This affected the recovery of NaCl 2, uranine 2, NaBr, and deuterium-labelled water that were applied upslope from this outflow. BDL stands for "below detection limit".**

| Location and tracer | Tracer recovery (% applied mass) | | | | | |
| --- | --- | --- | --- | --- | --- | --- |
| | After 100 minutes | | After 163 minutes | | After 24 hours | |
| | OF | TIF | OF | TIF | OF | TIF |
| *Clearing* | | | | | | |
| NaCl | 1 | 13 | 2 | 22 | 25 | 47 |
| Uranine | 25 | 2 | 26 | 3 | 17 | 3* |
| NaBr | <1 | <1 | <1 | <1 | 1 | <1 |
| $\delta^2H$ | 8 | 4 | 9 | 8 | 13 | 26 |
| *Grassland* | | | | | | |
| NaCl | 94 | 13 | No data | | | |
| Uranine | 97 | 7 | | | | |
| NaBr | BDL | BDL | | | | |
| $\delta^2H$ | 22 | 2 | | | | |

*Only for the first 385 minutes (i.e., 6.4 hours after application of first tracer line and 3.7 hours after the application of the second tracer line)

### 4.4. Overland flowpath lengths

Most of the dye infiltrated into the soil after a short distance, but there were a few longer flow pathways for both plots (Figure 8). The dye was frequently observed to exfiltrate a short distance below the initial infiltration point (Figure 8). In the clearing, the dye flowed over the surface for 0 to 0.8 m (mean: $0.5 \pm 0.3$ m) before infiltrating. There was a longer flowpaths on the right side of the plots (when looking upslope) that was 3 m long (up to 5 m when including the water that exfiltrated; Figures 8 and 9). In the grassland, flow pathways were generally longer and more similar to each other, typically between 0.5 and 2 m (mean: $1.1 \pm 0.3$ m). The exfiltration points were also closer to the infiltration points than in the clearing and appeared to occur on parallel lines (Figure 8). When including the exfiltrated water, the overall flowpath length was longer for the plot in the grassland (mean: $1.7 \pm 0.5$ m; Figure 9).

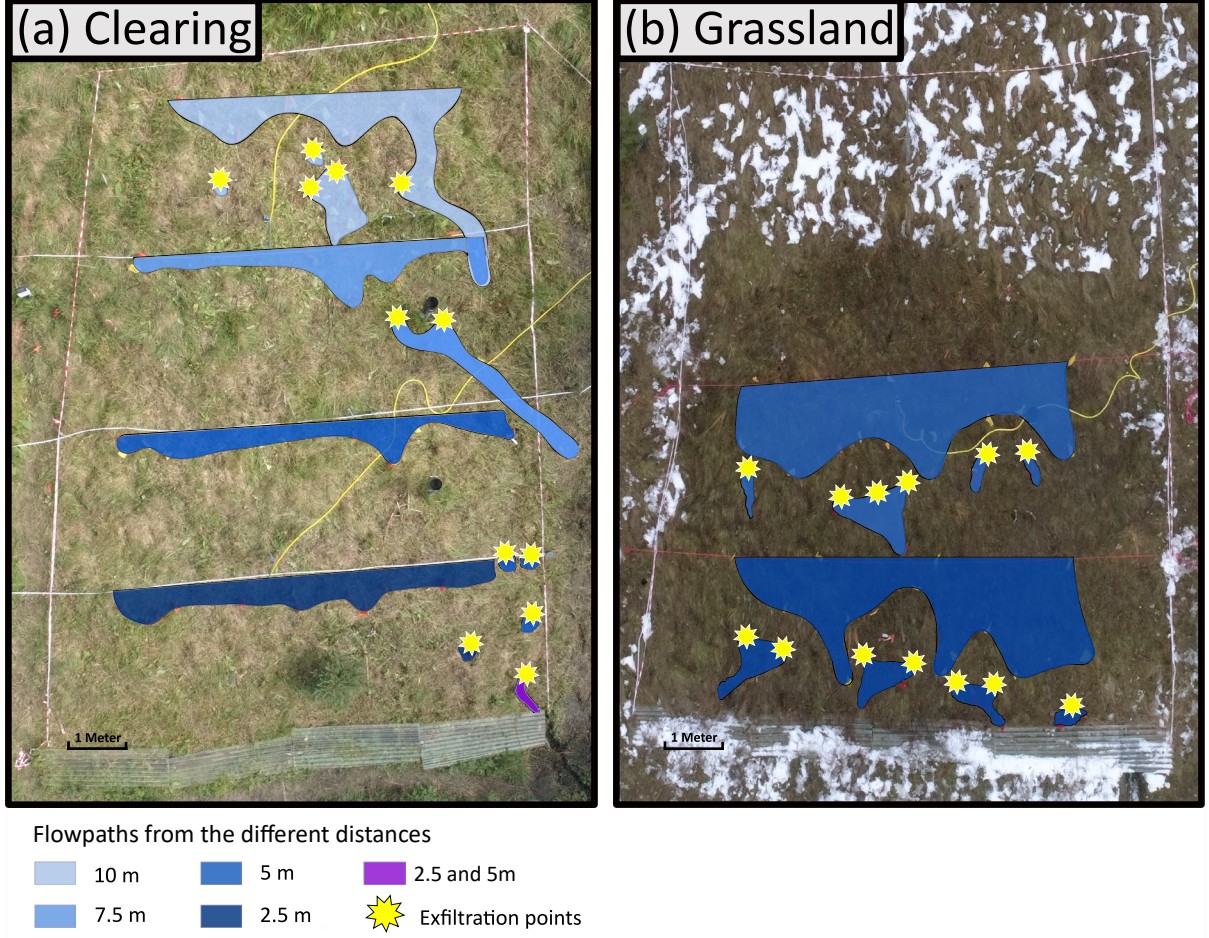

**Figure 8: Drone images of the plot the clearing (a) and the grassland (b), with polygons indicating the OF pathways on the surface (blue shading, with different colors representing the results for the dye applications at different distances from the trench), and the exfiltration points where the dye stained OF re-emerged at the surface (yellow stars). The violet shading indicates the flow path observed for the application at 2.5 m and at 5 m.**

500

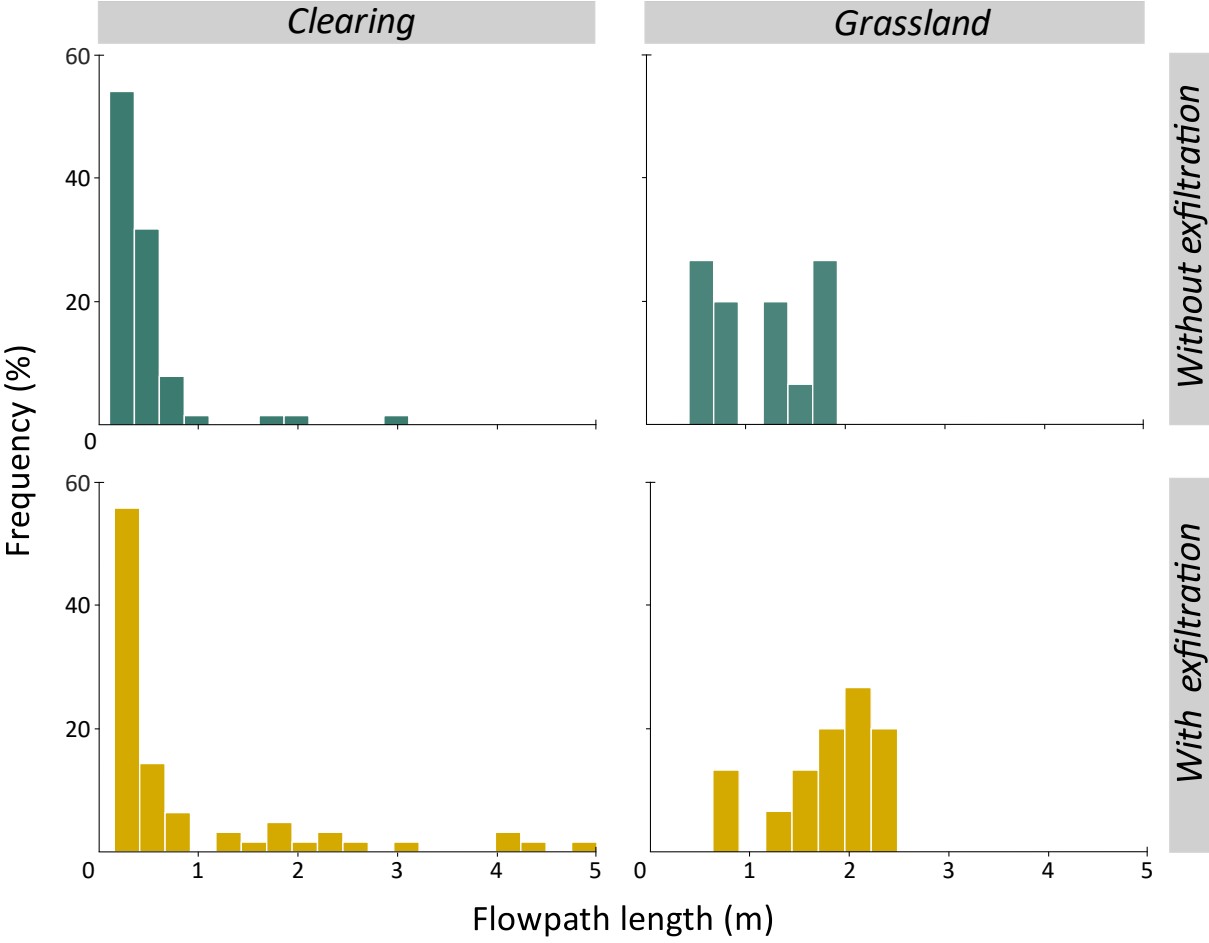

**Figure 9: Frequency distribution of the distances that OF travelled over the surface before infiltrating into the soil (top row), and the total flow path length when including the OF path of the exfiltrating dye stained water (bottom) for the experiments in the clearing (left) and the grassland (right).**

## 5. Discussion

### 5.1 Short OF flow pathways

The blue dye experiments showed that (as expected) OF does not occur over the entire plots. Instead, the OF pathways were short and OF infiltrated within several meters. However, the experiments also highlighted the importance of exfiltration after the OF travelled only a short distance below the surface. Return flow (RF; Dunne, 1978) appeared on the surface 0.5 to 5 m below the first infiltration point, and mostly created small ponds around the exfiltration points. Occasionally, the RF exfiltrated close to a depression or another exfiltration point, and the combined amount was enough to create a longer flow pathway. At both plots, the blue dye that exfiltrated still had an intense color, which suggests that it did not mix with a very large volume of soil water before exfiltration.

In the clearing, the OF paths were mainly short (mean: ~0.5 m), which would be expected in a forested environment with high conductivity soils (e.g., Gerke et al., 2015) but the microtopography also led to some longer flow pathways. This was mainly the case for the depressions that channelized OF on the upper part of the plot in the clearing to the middle of the plot, creating an OF flow pathway up to 3 m long (Figure 8a). The exfiltration points were randomly distributed across the surface of the plot in the clearing. A ~4 cm diameter pipe created a relative long subsurface flow pathway (up to 4.5 m), from which water exfiltrated just above the OF gutter (Figure 8a).

This pipe appeared to be a mouse burrow and was likely part of a macropore network as it provided almost all the OF collected in the gutter. Although other dye tracing studies for macropores (e.g., Mosley, 1979) have been criticised for creating unnatural boundary conditions, we do not think that the pouring of the dye on the surface caused the flow through these preferential flow pathways as we observed that exfiltration from the main "pipe" delivered the majority of OF during natural rainfall events as well. Pipes can reach up to tens of meters in some catchments (Jones, 2010, Wilson 2015) and can be major contributors to OF (called *pipe overland flow* by Putty and Prasad (2000)). Considering the runoff ratio of ~20% for OF for the plot in the clearing, it means that this macropore provided a considerable part of the total flow. Previous studies elsewhere (e.g., Tromp-van Meerveld and McDonnell, 2006; Uchida et al., 2003) have shown that single macropores can provide a very large portion of all the subsurface flow measured in a trench. The high celerity and velocity of OF and large amount of flow from this macropore (see also section 5.3.2) indicate that the network is quite shallow (i.e., near the surface), well connected, and that overland flow infiltrated quickly into it. If the network consists at least partly of mouse burrows, it is not surprising that it is very shallow because the water table is often located near the surface (Rinderer et al., 2014). Overall, these observations confirm those of previous studies that preferential flow via soil pipes and macropores is important for runoff generation in forested areas of the Swiss pre-Alps that are underlain by gleysols (Feyen et al., 1999; Weiler et al., 1998; Weiler and Naef, 2003).

In the grassland, the OF pathways were a bit longer than in the clearing and of a more similar length (~ 1 m), but also followed the microtopography. The exfiltration points were located closer to the infiltration points and were aligned across the plot. We attribute this to the microtopography caused by cattle trampling, where OF infiltrates on the top of the "terraces" caused by the trampling (and subsequent solifluction) and exfiltrates at the bottom (see Figure 8b).

Few studies have observed OF pathways in-situ. Dye is frequently used to observe flow pathways and macropores, but generally to visualize infiltration into the soil (Weiler and Flühler, 2004; Weiler and Naef, 2003) or lateral subsurface flow (Noguchi et al., 1999; Ehrhardt et al., 2022) and its connectivity (Anderson et al., 2009a). Only very few studies have used dye tracers to observe overland flow in natural environments (see Gerke et al., 2015; Maier et al., 2023). Nevertheless, the short OF pathways observed in this study agree with the general observation that OF pathways are short. For instance, Schneider et al. (2014) observed that dye applied to the surface on a 30° pre-Alpine grassland could flow 1 to 2 meters downslope from the application. Gerke et al. (2015) measured for a less vegetated forested slope flow paths up to 0.20 m in length. In a dryland, OF flowpath lengths were much longer, ranging between 1 to 6 m (Wolstenholme et al., 2020).

**5.2 Tracer transport**

The tracer experiments revealed the fast transport of solutes and the considerable interaction between OF and TIF, as the NaBr tracer applied to the subsurface of the plot in the clearing appeared after a short time in OF (Figure 6). As observed during the dye tracer experiments (section 5.1), this indicates the exfiltration of soil water (i.e., return flow, RF). Furthermore, the peak concentration of the NaBr tracer was higher for OF than for TIF, which suggests that some of the NaBr reached preferential flow pathways and exfiltrated as RF above the gutter, while the tracer that flowed through the subsurface mixed with a larger volume of water. However, the low recovery of NaBr suggests that most of the tracer remained in the subsurface (or left via the sides of the plot, or seeped into deeper soil layers). Feyen et al. (1999) showed that for a plot in a neighboring catchment, less than 0.5% of the stored soil water was mobilized during the event and that the majority of the soil does not contribute to flow in the subsurface.

This could suggest that a large part of the subsurface applied tracer could still be stored in the soil. However, the fact that a significant portion of the tracers applied to the surface of the clearing left the plot during the full day of sprinkling suggests that a substantial amount of tracers and water infiltrated into the soil matrix and was gradually released.

The heterogeneity of flow pathways, importance of infiltration of OF, transport through macropores and subsequent exfiltration as RF is highlighted by the recovery of the tracers that were applied to the surface of the clearing. The NaCl that was applied closest to the trench in the clearing was primarily recovered in TIF, suggesting predominantly infiltration of the tracer and subsurface transport. Contrary, the uranine applied 1.8 m further upslope was mostly recovered in OF (Figure 6). This suggests that this tracer infiltrated into the soil as well but also exfiltrated again after a short distance. In other words, the tracer applied further upslope reached a preferential flow network that provide RF, while the tracer applied further downslope did not and was routed mainly through the subsurface. Again, this highlights the high spatial variation and heterogeneity in flow pathways.

The influence of macropores and soil pipes in facilitating the transport of water and tracer was clearer for the plot in the clearing than the plot in the grassland. In the grassland, interactions between the flow pathways appeared less pronounced, as suggested by the predominant transport of the tracers in OF and much higher flow rate for OF than TIF (Figure 7). Furthermore, nearly all of the surface-applied tracers were almost fully recovered within 100 minutes because less of it infiltrated into the soil than for the plot in the clearing. This difference can be related to differences in the infiltration capacity related to differences in soil density and especially macroporosity. We observed a dense rooting system in the topsoil of the clearing but also found pieces of old buried wood deeper in the soil through (or along) which preferential flow might occur (see Noguchi et al., 1999). In the grassland, the roots were finer and denser at the surface than in the remainder of the topsoil, potentially limiting deeper infiltration and favoring biomat flow. The biomat flow was difficult to separate from pure overland flow, and therefore we refer to both as OF. However, the blue-dye experiments in the grassland still revealed the existence of exfiltration points (see Figure 8), which suggests that infiltration and exfiltration of OF (pure overland flow and biomat flow) takes place on the grassland as well. Furthermore, the recovery of the surface-applied tracers in TIF over the first 100-minutes was similar for both plots despite the much lower flow rates for TIF for the grassland plot. This suggests the occurrence of concentrated tracer transport through the subsurface in the grassland, while in the clearing, it must have mixed with a larger water volume, leading to a more diluted tracer concentration but overall comparable recovery.

The lack of any recovery of the NaBr tracer for the grassland plot might be related to the fewer number of macropores in the subsurface and overall slower flow through the subsurface (as also indicated by the low flow rates of TIF). Additionally, we might have applied too little water upslope of the tracer due to water supply limitations (see Figure 3). Alternatively, the lack of NaBr recovery might be due to the short time of the experiment on the grassland plot (although it was still considerably longer than the time required for the NaBr to arrive in the trench and gutter in the clearing).

Although the main differences in the flow rates and tracer transport between the plot in the clearing and the grassland can be attributed to the differences in the amount of TIF and number and size of the preferential flow pathways, other factors differed as well (Table 1). The grassland plot has a steeper slope (18°) than the clearing (9°). Infiltration is generally less for steeper slopes (Essig et al., 2009; Morbidelli et al., 2013), which can explain the low amount of TIF for the grassland plot and low tracer recover in TIF. The bulk density (Table 1) was higher

for the grassland than the clearing, which reflects the lower macroporosity and slower infiltration (Basset et al., 2023; Zhang et al., 2006). The higher bulk density may be the result of compaction due to cattle trampling (Hiltbrunner et al., 2012), less bioturbation in the grassland, or different rooting densities. The organic matter content near the surface (up to 15 cm) was also higher for the clearing, which may also explain the higher macroporosity (see Franzluebbers, 2001; Kochiieru et al., 2022).

Furthermore, there were differences in the experimental setup (Figure 3; Table 2). The higher recovery of the surface applied tracers for the grassland plot may be due to the higher rainfall intensity for the experiment on the grassland plot ($35 \pm 13$ vs $22 \pm 2$ mm h$^{-1}$). Higher intensity events can lead to more OF but may also lead to more preferential flow that can quickly transfer water and tracers through the topsoil. Feyen et al. (1999) did not measure any OF during lower intensity sprinkling experiments (8 mm h$^{-1}$) on a forested plot in a neighbouring catchment. Instead, the rainfall went primarily to deeper runoff and interflow through connected pores in the subsurface. Contrary, Weiler et al. (1999) conducted a high intensity sprinkling experiment (60 mm h$^{-1}$) on a forested plot in another nearby catchment and measured higher flow rates for OF than subsurface flow and high event water fractions ($f_e$) for both OF (90%) and subsurface flow (78%) (Weiler et al., 1999). However, measurements at 14 smaller plots by Gauthier et al. (2025) suggest that the precipitation thresholds for OF and TIF are similar and that the relative importance of OF to the total amount of near-surface flow (OF+TIF) increases with event size for grassland locations (i.e., with total precipitation).

Additionally, the experiments were done at different times of the year. As the experiment in the grassland was conducted at the end of the growing season, the vegetation was shorter and flattened by the snow that fell in the days preceding the experiments. This may have influenced the surface roughness and thus the OF dynamics (see Bond et al., 2020) and therefore the interaction between OF and subsurface flow pathways. We do not think that the snow cover itself affected the flow of the applied water as the initial testing of the sprinklers and the water applied to the plot to reach steady state flow conditions melted all the snow, and the soil temperatures were well above freezing (average 5.4 °C during the tracer experiments based on the temperature measured by the soil moisture sensors). The temperature of the water was similar (within 2°C) for OF and TIF at both plots, but it was much lower for the experiment on the grassland plot than the plot in the clearing (3°C vs. 12°C; Figure S2). This will have affected the kinematic viscosity and may have influenced the flow rates (see Schwab et al., 2016) and the particle velocity (Ni et al., 2019). It can thus also partly explain the lower velocities observed for the plot in the grassland than for the plot in the clearing.

The tracers used in this study (NaBr, NaCl, uranine, and deuterium) are all commonly used in hydrological tracer experiments because they are generally considered to be conservative and with minimal adsorption under typical soil conditions. In particular, the anions Br$^-$ and Cl$^-$ are well known for their low reactivity and high mobility in soils (e.g., Anderson et al., 2009b; Feyen et al., 1999; Scaini et al., 2017; Tsuboyama et al., 1994; van Verseveld et al., 2017). While we cannot entirely rule out small differences in transport behaviour among the tracers, we do not expect these to significantly influence the recovery for our experiments. Instead, the main reasons for the incomplete recovery (particularly for the plot in the clearing) are: 1) that a portion of the tracer likely remained in the soil, 2) lateral losses (as indicated by blue dye flow paths in Figure 8), 3) percolation into deeper soil layers or the bedrock that we did not capture with our collection systems, and 4) measurement uncertainties (e.g., in the flow rates).

### 5.3 High velocities and celerities

### 5.3.1. Velocities

The particle velocity (and celerity, see section 5.3.2) were high for both OF and TIF on both plots. The average (over all tracers) of the maximum particle velocity for OF was 51 m h$^{-1}$ for the plot in the clearing and 24 m h$^{-1}$ for the plot in the grassland. Few studies determined OF velocities for vegetated hillslopes but the values for the two plots in this study seem to be within the range of other studies, albeit on the lower side. For example, Holden et al. (2008) compared OF velocities on peatlands with different vegetation covers and bare surfaces and reported a mean overland flow velocity of 104 m h$^{-1}$ (range: 0.44-688 m h$^{-1}$). Bond et al. (2020) examined OF velocities on different grassland plots during different seasons in northern England by simulating an 18 mm h$^{-1}$ rain event. They measured OF velocities between 93 and 149 m h$^{-1}$. The lower velocities that we measured in this study could be due to differences in water application (inflow from sprinklers instead of hoses) and the relatively high flow rates in the other studies, leading to overland flow depths of up to 6 cm (Bond et al., 2020), which we did not observe. Furthermore, the plots in the previous studies were bounded and considerably smaller (0.5 m by 6 m (Holden et al., 2008) and 0.4 by 2.0 m (Bond et al., 2020)). Overland flow velocities might be higher on smaller, bounded plots due to limited infiltration opportunities, a reduced effect of microtopography, or edge effects. Additionally, the overland flow velocities reported by Holden et al. (2008) include measurements from bare plots where flow velocities were about 5-10 times faster than for vegetated plots. In our study, OF in the clearing was due largely to return flow from a soil pipe, which includes initial infiltration into the soil and subsurface transport that can reduce the particle velocity compared to the sheet flow observed in the other studies. In the grassland, sheet flow was visible, but short and the water infiltrated and then exfiltrated again, which will also have reduced the overall particle velocity compared to pure sheet overland flow.

The average (over all tracers) of the maximum particle velocity for TIF was 30 m h$^{-1}$ for the plot in the clearing and 17 m h$^{-1}$ for the plot in the grassland. These velocities fall within the upper range of velocities reported for forest and grassland sites with preferential flow pathways (see Anderson et al., 2009b; Wienhöfer et al., 2009), although higher velocities have been reported for pipeflow and macropore flow (e.g., Graham et al., 2010; Mosley, 1979, 1982). For instance, Feyen et al., (1999) conducted tracer experiments on two 13 m$^2$ forested plots in a neighboring catchment with a sprinkling rate of ~8 mm h$^{-1}$ and applied bromide as a line tracer to the surface. The calculated velocities during this experiment were 9 m h$^{-1}$ for the site with muck humus and 0.5 m h$^{-1}$ for the site with mor humus. The velocity of a salt tracer that was injected into the topsoil at 30 cm depth was 11 m h$^{-1}$ for the muck humus site and 3 m h$^{-1}$ for the mor humus site. These findings highlighted that fast flow through the topsoil via a network of conducting pores. Weiler et al. (1998) conducted high-intensity (60-100 mm h$^{-1}$) sprinkling experiments in a near-by catchment on somewhat similar sized plots located in a grassland and a forest and reported higher flow velocities for the grassland (22-144 m h$^{-1}$) than the forest (5.4 m h$^{-1}$). The higher flow velocities in the grassland were attributed to macropores created by animals, which were larger in diameter compared to the denser but smaller pores formed by plant roots in the forest. For the plot in the clearing in this study, we observed several macropores created by animals, thus the plot is more comparable to the one in the grassland of Weiler et al. (1998). In the grassland plot in this study, the compaction by cattle trampling likely reduced the infiltration by reducing the size of the macropores, leading to slower vertical and lateral particle transport, and therefore slower velocities for the grassland plot than in the clearing.

### 5.3.2. Celerities

The average value of the celerities for OF (over all locations where we applied the water pulse) was 150 m h$^{-1}$ for the plot in the clearing and 64 m h$^{-1}$ for the plot in the grassland. In the clearing, we found a particularly high celerity for OF for the pulse at 4 m from the trench and gutter (240 m h$^{-1}$). After removing this outlier, the mean celerity of OF in the clearing was still high (105 m h$^{-1}$) and almost twice as high as for the grassland. The celerities for TIF were similar for the clearing (34 m h$^{-1}$) and the grassland (41 m h$^{-1}$).

The near saturated, steady state conditions in our study make it difficult to directly compare the celerities with those reported in other studies. In many other cases, the celerity estimates account for vertical flow through the unsaturated zone and changes in soil water storage during wetting. In contrast, our experiments were conducted under near saturated conditions where storage changes were minimal. Furthermore, there are differences in the way that celerities are calculated. Still, the celerities for TIF are lower than the initial hillslope celerities reported by Scaini et al. (2018) for natural events for a catchment in Luxembourg ($90 \pm 106$ m h$^{-1}$) but are comparable to their integrated hillslope celerities ($25 \pm 34$ m h$^{-1}$) for flow at the soil-bedrock interface, which is much deeper than the flow pathways studied here. The celerities found in our study are much higher than those reported by van Verseveld et al. (2017) (0.01-0.4 m h$^{-1}$), who estimated wetting front celerities during sprinkling experiments based on soil moisture, water level and soil matric potential measurements. However, the average sprinkling rate used by van Verseveld et al. (2017) was only 3-4 mm h$^{-1}$ and the experiment was conducted on unsaturated soils.

The high celerity for OF for the plot in the clearing resulted in a much larger difference between the celerity of OF and TIF in the clearing than for the grassland. We hypothesize that the high celerity for OF in the clearing is due to flow through almost filled soil pipes that lead to return flow (RF) just above the trench and gutter (see also section 5.1). The high sprinkling intensities and near saturated conditions at steady state conditions during the experiment, likely contributed to the high celerities (and also high velocities). However, these near saturated conditions are not uncommon in the Studibach catchment (see also Gauthier et al., 2025; Rinderer et al., 2014), as reflected by the frequent occurrence of groundwater levels near the surface, the return period of the applied rainfall intensity and the relatively short wetting period required to reach saturation/steady state conditions during the experiments.

### 5.3.3 Comparison of velocities and celerities

Previous studies compared the velocity and celerity by using the kinematic ratio ($\alpha_k$ being the ratio of celerity divided by velocity; Rasmussen 2000). For our study, $\alpha_k$ for OF ranged between 2 and 3. For TIF, this ratio depended on the location. For the plot in the grassland, the celerity was also two to three times higher than the particle velocity but for the plot in the clearing, the celerity and particle velocity were nearly identical. The latter is similar to the values reported by Scaini et al. (2017) ($\alpha_k$ :1.02-1.06) for subsurface flow in a permeable soil above a slate bedrock in a forested catchment in Luxembourg. Contrary to our study and the study by Scaini et al. (2017), van Verseveld et al. (2017) and Torres et al. (1998) found much higher values of $\alpha_k$ for high permeability colluvial soils (3-20 and 15, respectively). For flow through discrete pore networks (rather than more diffusive flow through the soil matrix) $\alpha_k$ tends toward one (Hrachowitz et al., 2016). The results thus provide further evidence of the importance of preferential flow, especially in the clearing, as also highlighted by the blue dye experiments (section 5.1) and the tracer recovery (section 5.2).

## 6. Conclusions

We used rainfall simulation and tracer experiments on two 8 m wide trenched plots in a steep humid pre-Alpine catchment with low permeability gleysols to better understand overland flow (OF) and lateral flow through the topsoil (topsoil interflow, TIF). For the plot in a natural clearing in the open forest, the applied water infiltrated quickly into the soil and was routed through preferential flow networks (i.e. macropores) downslope. Part of this water resurfaced as return flow a few meters after infiltrating, but most left the plot as lateral flow through the topsoil. For the plot in the grassland, most of the applied water left the plot as OF (including biomat flow), and less water was transported as TIF, likely due to the lower macroporosity of the soil. Tracer transport during steady state flow conditions was fast, with velocities ranging between 17 to 51 mm h$^{-1}$. The celerity was 2-3 times higher than the velocity, except for TIF in the clearing for which it was similar as the velocity. The celerity and velocity of OF were higher for the plot in the clearing than the plot in the grassland, and were always higher than for TIF. The celerity for TIF was similar between the two plots, but the velocity of TIF was higher for the plot in the clearing than the plot in the grassland. The differences in celerity and velocity for the two plots are mainly attributed to the difference in the preferential flow networks, flowpath lengths, and shape. Together, these findings highlight the importance of preferential flow for the fast response of OF and TIF, and likely also the fast response of streams in this (and other) pre-Alpine catchments underlain by gleysols.

## Data availability

Data can be provided by the corresponding authors upon request or can be accessed from the EnviDat.ch repository (https://doi.org/10.16904/envidat.685, Gauthier and Leuteritz, 2025).

## Author contributions

AL, VAG and IvM conceptualized the study and planned the experiments and data collection. AL and VAG collected the data, analyzed the data and wrote the manuscript draft; AL, VAG, and IvM reviewed and edited the manuscript. IvM supervised the project. The order of the two first authors was determined with a coin flip (with a 2 CHF coin from 1973 on the blue square at Irchel campus on 3.4.2025).

## Competing interests

The authors declare that they have no conflict of interest.

## Acknowledgements

We thank our colleagues from the H2K group at the Department of Geography at the University of Zurich, Elena Köpfli, Anja Ehrensperger, Sandro Wiesendanger, Amaury Berjaoui, and Louise Fuchs for their help in setting up the plots and the experiments. We thank Barbara Herbstritt (University of Freiburg) for the isotope analyses and Björn Studer (ETH Zurich) for the bromide analyses. We thank the Oberallmeindkorporation Schwyz (OAK), the Department of Environment of the Canton of Schwyz, and the Alpthal municipality for their cooperation. This

research was conducted as part of the TopFlow: (in)visible water flows near the surface project funded by the Swiss National Science Foundation (Grant 197194).

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
