# Peer review of "Tracing near-surface runoff in a pre-Alpine headwater catchment"

_EGUsphere, 2025_

## Author Comment (AC1)

Dear Jan Wienhöfer,

We thank you for this positive feedback, the careful review of our manuscript and the useful comments. We will use these to improve the manuscript. We respond to each comment below (original comment in black font, our response in blue font).

The manuscript "Tracing near-surface runoff in a pre-Alpine headwater catchment" investigates overland flow and interflow dynamics at two Alpine sites using sprinkler experiments and different tracers. Overall, this is a fascinating and highly valuable study. The authors have clearly dedicated substantial effort to producing a unique and rich set of qualitative and quantitative observations, which significantly advance our understanding of runoff generation in Alpine catchments.

The authors' work is a significant contribution to the field, and the manuscript is comprehensive and well written. While reading the draft, however, I noted a few points that could benefit from further clarification or refinement. These are outlined below and, once addressed, will help strengthen the manuscript for publication in HESS.

**Specific Comments**

Page 3, Line 101-104: The claim seems to be that fast flow pathways are a key factor because streamflow reacted earlier than groundwater in 'about half of the events'. This raises the question if that is special compared to other catchments, or is this typical? And how does this relate to velocity (fast flow paths) vs celerity (earlier rise)?

We thank the reviewer for this question. A quicker rise in streamflow compared to groundwater levels is indeed observed in many catchments and is not unique to our study area (e.g., Beiter et al., 2020). Generally, this is seen as an indication that other processes than groundwater flow or subsurface flow are responsible for the initial increase in streamflow during an event, such as precipitation falling on the channel or overland flow on near stream areas.

This observation refers to celerity, i.e., the speed at which a change in pressure or water level propagates, rather than velocity, which refers to the movement of water particles or solutes. Since we are discussing timing of hydrograph responses (rather than tracer movement), we will rewrite this sentence for greater clarity.

Page 6, Line 185-187: Do these statistics still hold when taking the duration of rainfall into account? If the return period of 24 mm/h over one hour is 1 to 2 years, the return period of a rainfall of 24 mm/h over 24 hours will be much higher. It would be helpful if you could include this information and discuss the possible implications.

As the sprinkling experiments were relatively short, we decided to use a similar event duration and intensity to estimate the return period. Nevertheless, if we take into account the same rainfall intensity for longer event duration, the return period will increase. A 38 mm event in 24 hours occurs on average

11 times/year. For a 24 mm event it is 24 times/year.  We will add this additional information and your suggested comparison of the experiment for events with longer duration in the revised manuscript.

Page 10, Line 259: Please be more specific: how were the samples selected? Were not all samples analysed for all tracers?

Only for the Bromide and Deuterium were the samples analysed in the lab. Uranine concentrations and EC were measured continuously in the field using sensors at 1-minute intervals. Not all collected samples were analysed due to financial and time constraints. During the tracer experiments, we intentionally oversampled the breakthrough curve by collecting samples at one-minute to two-minute intervals for the first hour of sampling after tracer application (see Table 1). We analysed only part of the samples to determine whether a clear tracer breakthrough could be identified. As shown in Figures 6g and S1c-d, the breakthrough curves could be very well determined by these samples already. Analysis of the additional samples collected in between the selected samples would not have provided significant new information on the breakthrough curves.

We account for any discrepancies in the time of the arrival (1 minute max if we did not analyse the absolute first sample with an elevated concentration due to sample selection) in the 2-minute uncertainty used in the velocity calculations. In Table 1, we show more clearly which samples were selected for laboratory analysis.

*Table 1: Overview of the samples of overland flow and topsoil interflow that were analysed for Bromide and Deuterium for the tracer experiments in the clearing and the grassland. The start of tracer application is indicated with pink color.*

| Clearing | | | Grassland | | |
|---|---|---|---|---|---|
| Date and time | Overland flow | Topsoil interflow | Date and time | Overland flow | Topsoil interflow |
| 2023-08-09 9:47 | | x | 2023-11-08 14:04 | x | x |
| 2023-08-09 9:58 | x | | 2023-11-08 14:20 | x | x |
| 2023-08-09 10:05 | | x | 2023-11-08 14:21 | x | |
| 2023-08-09 10:12 | x | | 2023-11-08 14:22 | x | x |
| 2023-08-09 12:02 | x | x | 2023-11-08 14:23 | x | |
| 2023-08-09 12:07 | x | x | 2023-11-08 14:24 | x | x |
| 2023-08-09 12:10 | x | x | 2023-11-08 14:25 | x | |
| 2023-08-09 12:11 | x | | 2023-11-08 14:26 | x | x |
| 2023-08-09 12:12 | x | x | 2023-11-08 14:27 | x | |
| 2023-08-09 12:13 | x | | 2023-11-08 14:28 | x | x |
| 2023-08-09 12:14 | x | x | 2023-11-08 14:29 | x | |
| 2023-08-09 12:15 | x | | 2023-11-08 14:30 | x | x |
| 2023-08-09 12:16 | x | x | 2023-11-08 14:32 | x | x |
| 2023-08-09 12:17 | x | | 2023-11-08 14:34 | x | x |
| 2023-08-09 12:18 | x | x | 2023-11-08 14:36 | x | x |
| 2023-08-09 12:19 | x | | 2023-11-08 14:38 | x | x |
| 2023-08-09 12:20 | x | x | 2023-11-08 14:40 | x | x |

| | | | | | |
|---|---|---|---|---|---|
| 2023-08-09 12:22 | x | x | 2023-11-08 14:42 | x | x |
| 2023-08-09 12:24 | x | x | 2023-11-08 14:44 | x | x |
| 2023-08-09 12:26 | x | x | 2023-11-08 14:46 | x | x |
| 2023-08-09 12:28 | x | x | 2023-11-08 14:48 | | x |
| 2023-08-09 12:30 | x | x | 2023-11-08 14:50 | x | x |
| 2023-08-09 12:32 | x | x | 2023-11-08 14:54 | x | x |
| 2023-08-09 12:34 | x | x | 2023-11-08 14:58 | x | x |
| 2023-08-09 12:36 | x | x | 2023-11-08 15:02 | x | |
| 2023-08-09 12:38 | | x | 2023-11-08 15:06 | x | x |
| 2023-08-09 12:40 | x | x | 2023-11-08 15:10 | x | x |
| 2023-08-09 12:44 | x | x | 2023-11-08 15:14 | x | x |
| 2023-08-09 12:48 | x | x | 2023-11-08 15:18 | x | x |
| 2023-08-09 12:52 | x | x | 2023-11-08 15:20 | x | |
| 2023-08-09 12:56 | x | x | 2023-11-08 15:22 | | x |
| 2023-08-09 13:00 | x | x | 2023-11-08 15:26 | | x |
| 2023-08-09 13:04 | | x | 2023-11-08 15:30 | x | x |
| 2023-08-09 13:08 | | x | 2023-11-08 15:35 | | x |
| 2023-08-09 13:10 | x | | 2023-11-08 15:40 | x | x |
| 2023-08-09 13:12 | | x | 2023-11-08 15:45 | | x |
| 2023-08-09 13:16 | | x | 2023-11-08 15:50 | x | x |
| 2023-08-09 13:20 | x | x | 2023-11-08 16:00 | x | x |
| 2023-08-09 13:25 | | x | 2023-11-08 16:10 | x | x |
| 2023-08-09 13:30 | x | x | 2023-11-08 16:20 | x | x |
| 2023-08-09 13:35 | | x | 2023-11-08 16:30 | x | x |
| 2023-08-09 13:40 | x | x | | | |
| 2023-08-09 13:50 | x | x | | | |
| 2023-08-09 14:00 | x | x | | | |
| 2023-08-09 14:10 | | x | | | |
| 2023-08-09 14:15 | x | | | | |
| 2023-08-09 14:20 | | x | | | |
| 2023-08-09 14:30 | x | x | | | |
| 2023-08-09 14:40 | | x | | | |
| 2023-08-09 14:45 | x | | | | |
| 2023-08-09 14:50 | | x | | | |
| 2023-08-09 15:00 | x | x | | | |
| 2023-08-09 15:10 | | x | | | |
| 2023-08-09 15:20 | | x | | | |
| 2023-08-09 15:30 | x | x | | | |
| 2023-08-09 15:40 | | x | | | |
| 2023-08-09 15:45 | | x | | | |
| 2023-08-09 16:00 | x | x | | | |

| | | | | | |
|---|---|---|---|---|---|
| 2023-08-09 16:15 | | x | | | |
| 2023-08-09 16:30 | x | x | | | |
| 2023-08-09 16:45 | | x | | | |
| 2023-08-09 17:00 | x | x | | | |
| 2023-08-09 17:25 | | x | | | |
| 2023-08-09 17:35 | x | | | | |
| 2023-08-09 17:55 | | x | | | |
| 2023-08-09 18:05 | x | | | | |
| 2023-08-09 18:15 | | x | | | |
| 2023-08-09 18:48 | | x | | | |
| 2023-08-09 19:00 | x | | | | |
| 2023-08-09 21:00 | x | | | | |
| 2023-08-09 23:00 | x | | | | |
| 2023-08-10 1:00 | x | | | | |
| 2023-08-10 3:00 | x | | | | |
| 2023-08-10 5:00 | x | | | | |
| 2023-08-10 7:00 | x | | | | |
| 2023-08-10 9:00 | x | | | | |
| 2023-08-10 10:50 | x | x | | | |
| 2023-08-10 11:50 | | x | | | |
| 2023-08-10 13:17 | | x | | | |

Page 12, Line 341: where did the remaining third of the water go?

The runoff ratio was 20% for OF and 46% for TIF, indicating that we collected 66% of the water applied to the plot at the trench during the experiments. The remaining one third of the water might have either percolated in deeper soil layers or flowed laterally out of the plot as they were not bound (e.g. via the "blue dye" channel in clearing leaving the plot on the right-side Fig.8.a). We will add some explanation about this in the revised manuscript.

Page 15, Line 398: NaBr was applied to shallow piezometers. Did you observe any overflowing of these boreholes?

We did not observe any overflow from the tubes.

Page 17, Table 5: Why are the velocity estimates from the Deuterium results not included here?

We did not include Deuterium in the velocity estimates as we applied the Deuterium-labelled water diffusely via the sprinklers. The application distance to the trench ranged from a few centimeters to several meters, which makes it difficult to define a single representative travel distance to calculate the particle velocity. Although we observed Deuterium breakthrough at the trenches, the uncertainty in the

travel distance makes the derived velocity estimate not directly comparable to the tracers applied in a line at a defined distance from the trench. For this reason, we chose not to include Deuterium in Table 5.

However, based on the observed first arrival times of deuterium at the trench and estimated travel distances between 0.5-10 m in the clearing and 0.5-7.5 m in the grassland, we can estimate a possible maximum velocity range of 8-150 m h$^{-1}$ for OF and 3-50 m h$^{-1}$ for TIF in the clearing and 10-150 m h$^{-1}$ for OF and 4-56 m h$^{-1}$ for TIF in the grassland.

Page 18, Line 445-454: Could you elaborate more on this topic? What does the incomplete recovery mean for your conclusions? Are the differences in tracer transport and recovery only because of the nature of the different flow paths, or are properties of the compounds also an explanation that needs to be considered. e. g, different adsorption characteristics?

The tracers used in this study (NaBr, NaCl, Uranine, and Deuterium) are all commonly used in hydrological tracer experiments because they are generally considered to be conservative and exhibit minimal adsorption under typical soil conditions. In particular, the anions Br$^-$ and Cl$^-$, as well as Deuterium, are well known for their low reactivity and high mobility in soils (e.g., Anderson et al., 2009; Feyen et al., 1999; Scaini et al., 2017; Tsuboyama et al., 1994; van Verseveld et al., 2017). While we cannot entirely rule out small differences in transport behavior among the tracers, we do not expect these to significantly influence the recovery for our experiments. The main reasons for incomplete recovery are: 1) a portion of the tracer likely remained in the soil, 2) lateral losses (as indicated by blue dye flow paths in Figure 8), 3) percolation into deeper flow paths that we could not capture in our collection systems, and 4) measurement uncertainties (e.g., in the flow rates). We will clarify this in the revised manuscript.

Page 19, Table 6: Unfortunately, it is not fully clear what is shown here exactly. Is the recovery expressed cumulatively? Is it the percentage of the total applied mass, or only for the first tracer lines in the first columns? Please add explanation to the table caption. Maybe consider moving the last two sentences from the caption to the discussion.

Table 6 shows the cumulative tracer recovery as percentage of the applied mass of the respective tracer. We chose to compare the tracer recovery after 100 minutes as the tracer experiment in the grassland was limited to 100 minutes. The tracer experiment in the clearing could be conducted for 24 hours. We included the recovery after 163 minutes for the experiment in the clearing as this was the time when we applied the second set of tracer lines for NaCl and Uranine. The recoveries after 24 hours include therefore the recovery for both sets of tracer lines.

Page 22, Line 564: Would this not require a similar velocity of transport at both sites to be a fair comparison?

We thank the reviewer for this question. Our statement refers to the total mass of tracer recovered in TIF over the first 100 minutes of the experiments. This comparison focuses on the overall tracer recovery integrated over the 100-minute period. Of course, very different velocities would lead to very different

tracer recoveries over a certain period but in this case the focus is on recovery and therefore the comparison is still valid. In fact, the point we aimed to emphasize here is that, despite lower TIF flow rates in the grassland (and higher contributions from overland flow), the amount of tracer reaching the TIF outlet was still similar to that in the clearing. This suggests that there is concentrated tracer transport through the subsurface occurring at the grassland plot. We will clarify this point in the revised manuscript.

Page 23, Line 594 - 601: how about ambient temperature? Water viscosity changes a lot with temperature, and that influences the flow velocity.

This is an interesting aspect. We will acknowledge that the viscosity was different for the two experiments in the revised text.

The soil temperature was different between the two locations, as one experiment was carried out in August for the clearing and the other in November for the grassland. The temperature of the water was similar (+/- 2 °C) for OF and TIF (which has little influence on the viscosity) at each location. For the experiment in the clearing the average temperature of OF and TIF was 12 °C and for the experiment in in the grassland, it was 3 °C, which gives respectively kinematic viscosity of 1.23 and 1.64 mPa/s. The following figure represents the temperature of the water that was collected at the trench (not the soil water directly).

[Figure]

*Figure 1: Time series of the OF (blue) and TIF (orange) flow rates (L min-1) during the tracer experiments on the plot in the clearing (a) and the plot in the grassland (b). The associated temperatures are in dark blue for OF and orange for TIF (C°).*

To have an estimation of the influence of the viscosity on the velocity we can use the following Poiseuille law (for laminar flow):

$$v_{max} = \frac{R^2}{2\mu}\left|\frac{dp}{dz}\right| \text{ (for two dimensions)}$$

where **R** is the radius of the tube and **μ** the viscosity. If we take a similar tube size and condition, the velocity would be 33% slower, due to the lower viscosity in the grassland than in the clearing. However, this result is valid only for laminar flow, which may not be the case for preferential flow though soil

pipes. In the revised discussion, we will point out the difference in water temperature and how it could impact the velocity as well.

Page 25, Line 604-635: This part of the discussion could be more elaborate. The OF velocity will be determined by slope and surface characteristics (roughness, infiltrability), which in turn will be determined by the types and states of vegetation and soil. Also, the temperature (viscosity of the water) and other experimental conditions like length of the flow paths also play a role. It is thus not only 'vegetated' vs 'bare' soil. Comparing mean (see below) velocities without normalizing for these factors is not really conclusive.

Thank you for this comment. We will reformulate and add information between line 604 and 635 to the revised manuscript. The comparison between bare and vegetated soil refers to the study of Holden et al. (2008). We will be clearer and more careful about this reference in the revised manuscript.

For OF, the velocity was higher in the clearing than in the grassland. In the clearing, most of the collected OF came from a large and shallow pipe flow. The infiltration rate was high in the clearing as well as we almost did not observe any sheet overland flow. Therefore, fast infiltration and fast flow through the pipe have led to this higher velocity. Contrary, in the grassland the infiltration rate was low, triggering more sheet-like overland flow. The surface of the soil was well vegetated and probably reduced the velocity of the flow due to friction. For TIF, the velocities are higher in the clearing than in the grassland, which might be due to soil compaction or fewer larger roots in the grassland. This compaction which reduced the infiltration in the grassland, can also reduce the size of the macropores, and lead to slower particle propagation in smaller macropores and therefore slower velocities in the grassland than in the clearing.

Following your previous comment, we will add also information about the viscosity and its impact on the velocity that was measured. Also, we will add the information about how local factors influence the velocity between the grassland and the clearing.

How were the velocities averaged? Arithmetic or harmonic mean? This also applies to the other average velocities reported here. Example: When the time that overland flow needs to travel a distance of 2 m would be 1 minute and 2 minutes, the average velocity would be 1.3333 m per minute (harmonic mean).

We thank the reviewer for this question. We report arithmetic mean values of the velocities in the manuscript. We chose the arithmetic mean because we treat each experiment (each line of tracer or water pulse) as an independent test rather than a cumulative transport scenario. Additionally, to account for the influence of the main outlier (4-meter line in the celerity experiment in the clearing) we give the mean values with and without this particular event included) (see Lines 638-641). The individual celerity and velocity values for each experiment are given in Table 4 and Table 5, so that readers can calculate the harmonic mean if desired.

How would the measured flow velocities compare with theoretical estimates, e.g., Gauckler-Manning-Strickler formula? Would you get realistic roughness values when inverting the formula?

The processes were different between the OF and TIF. Therefore, it would be important to use different equations. For propagation into the soil for TIF, we would probably use the Darcy equations, but we lack information about the soil characteristics (e.g. hydraulic conductivity) and flow was likely preferential. Furthermore, it would be important to implement two components, vertical infiltration and lateral propagation.

Only in the grassland can the overland flow processes be approximated with the Manning-Strickler formula:

$$K = \frac{v}{i^{1/2} * R^{2/3}}$$

Where **K** is the Strickler coefficient (inverse of Manning's coefficient **n**), **v** the velocity, **i** the slope and **R** the hydraulic radius. If we take the slope of 32.5%, hydraulic radius of 0.0298 $m^2$ (8 m long on 0.03 m depth (including the biomat flow as we collecting it) and the average velocity calculated of 0.018 m/s (measured). It gives us a value of **K** = 0.033 for the Strickler coefficient, which is out of the range of Chow (1959). Masoodi & Kraft (2024) suggest that the Manning coefficient should be adapted for overland flow when there is vegetation and suggest using a linear method (Hinsberger et al., 2022; Oberle et al., 2021) for different heights of vegetations. In our case, overland flow is below the vegetation a close to 0 (ground level), which would give a transformation from Strickler to Manning coefficient as follow:

$$n = \frac{K}{5}$$

The manning coefficient would be then **n** = 0.0066.

Unfortunately, this result is far from the typical ranges for Manning coefficients, which suggests that other processes are important in our setup. One of the possible explanations is that that the overland flow that reached the gutter came only from a small area as overland sheet flow (see figure 8b). Therefore, most of the overland flow was collected as biomat flow, and thus after infiltration, which would fall in the domain of subsurface lateral flow.
If we only use 3mm as the depth of the overland flow (excluding biomat flow) for the hydraulic radius, **n** = 0.03, which would fall in a range between smooth surface and fallow (Engman, 1986).

To obtain more accurate values for the Manning's coefficient, the experiment should have been done in a different setup such as Holden et al. (2008) to create more sheet flow.

Page 26, Line 685: The data should be uploaded before publication. In fact, it would be helpful if they could be included in the review. Otherwise, chances are too high that this will never happen.

We agree that uploading the dataset before the publication is useful. We will upload the data to the WSL repository envidat.ch and provide a link to the data in the revised manuscript.

**Minor Comments / Clarifications**

Page 3, Line 103: What does 'close' mean, in m?

The groundwater varies between 1.5 m deep to the soil surface in the catchment. In our locations, this variation is smaller and varies from 1 m to the surface. We will add this information to the revised manuscript.

Page 5, Line 125-127: Please be a bit clearer: 10 cm organic rich AND another horizon rich in organic with 30 cm thickness, or up to 30 cm depth? Would that be A and B horizon, or litter layer and A horizon, or something else?

The soil is a mollic gleysol with a muck humus layer (which is considered as the A horizon, 0 to 30 cm depth) underlain by a reduced Bg horizon (clay) below. Within the A horizon, we could differentiate between a 10 cm upper layer and a 20 cm lower layer based on the degree of decomposition and roots density. We will add a clearer explanation to the revised manuscript.

Page 5, Line 128: Figure S1 is not about roots

Thank you for pointing this out. It was a mistake as we removed the soil profiles from the supplementary material. We will make sure that every reference to a figure or a table in the text is correct.

Page 8, Table 3: Tracer volumes are given for Uranine and Deuterium - what were the masses? Please specify to align with the table header

We thank the reviewer for this comment. We will provide the applied masses of tracers to the revised manuscript.

Page 18, Figure 7: Is this the from the Deuterium experiment? Please add more info to the caption.

Yes, this figure shows the flow rates of overland flow and topsoil interflow in the clearing and grassland plot and the darker shades of blue and orange indicate the contribution of deuterium-labelled water. We will clarify this in the revised manuscript.

Page 20, Fig 8: These are great images. Perhaps make them a bit larger (page width)?

Thank you, we will enlarge them to a full-page size in the revised manuscript.

Page 22, Line 543: Does this refer to Deuterium labeled water? Please clarify.

Thank you for this comment. We refer to all surface-applied tracers in this sentence. We will clarify this in the revised manuscript.

Page 22, Line 562: That could possibly be exfiltration from biomat flow, right?

It could be yes. It is difficult to differentiate visually between OF as sheet flow and the biomat flow. In this case, most likely the water that exfiltrated was a mixture of pure OF and biomat flow. In our case, the biomat was acting as the shallowest saturated layer, and then a sheet OF occurred on top of it. Our collecting system does not permit us to differentiate between the processes as we collected the flow between the surface to 3 cm depth. We will add information about the processes and highlight that we collected both OF and biomat flow in the revised manuscript.

Page 22, Line 564 – Page 23, Line 566: This requires a little more explanation. Would that mean that concentrations were much higher with the lower flow rates?

Indeed, the concentrations of NaCl and Uranine were higher in TIF in the grassland compared to the clearing. Therefore, despite lower TIF flow rates in the grassland, the total tracer mass recovered in TIF over the first 100 minutes was similar between the two plots. This indicates that less water, but with higher tracer concentration, was transported as TIF in the grassland, while in the clearing, a larger water volume with more diluted tracer concentration (due to more mixing with non-labelled soil water) resulted in a comparable overall recovery. We will clarify this in the revised manuscript.

Page 24, Line 609: What does this flow rate should tell the reader? Isn't the surface area/wetted perimeter equally important?

Indeed, the wetted surface area would also be important and the flow rates given here are not informative. The cited study by Holden et al. (2008) used small, bounded plots (0.5 m wide and 6 m in length) with different vegetation covers and bare soil and applied water with varying flow rates (0.05 – 0.5 L s$^{-1}$) to the plot via a variable-speed pump and measured travel times of Rhodamine WT dye after flow had become constant at the plot outlet.

We realize that this additional information is not important for the point that we want to make and instead of giving more information, we will rewrite and shorten this sentence to avoid confusion.

Page 25, Line 642: Please clarify why the saturated and steady-state conditions would make comparisons difficult?

In the other studies, the authors need to account for vertical flow through the unsaturated zone and especially the change in storage in the soil during soil wetting. In contrast, our study was conducted during near-saturated, steady-state conditions where storage changes were minimal. We will rewrite the sentence and add these arguments for greater clarity.

Page 25, Line 673: info that these are 'trenched' maybe more important than the width

Thank you. We will change this sentence in the revised manuscript and mention trench in addition to the 8 m wide plots. We would like to keep the 8m wide here to indicate that the plots were relatively large.

Page 26, Line 680: maybe include a comment on the difference in OF and TIF velocities - both are fast, but also OF still is significantly faster

Thank you, it is indeed one of our important findings that we forgot to mention in the conclusion. We will rewrite the last sentences and add this information to the revised manuscript.

**Technical Corrections**

Thank you for the following technical corrections. We will correct and upload the following mistakes to adapt the manuscript to the HESS requirements and make sure that there are no other errors in the updated manuscript.

Page 6, Line 160: "(see Gauthier et al. (2025))" - Consider avoiding the double parentheses - check style guide

Page 10, Line 283: "h" -  variables are set in italic, please check style guide – also variables elsewhere

Page 10, Line 270: "containing the 3 mg L-1 brilliant blue dye" – check wording/sentence structure

Page 10, Line 272: "tree" - typo

Page 10, Line 279: "was able to see" - check wording/sentence structure

Page 18, Line 450: "large" - Please check

Page 25, Line 651: Check sentence structure

**References:**

Anderson, A. E., Weiler, M., Alila, Y., & Hudson, R. O. (2009). Subsurface flow velocities in a hillslope with lateral preferential flow. Water Resources Research, 45(11). https://doi.org/10.1029/2008WR007121

Beiter, D., Weiler, M., & Blume, T. (2020). Characterising hillslope–stream connectivity with a joint event analysis of stream and groundwater levels. Hydrology and Earth System Sciences, 24(12), 5713–5744. https://doi.org/10.5194/hess-24-5713-2020

Chow, V. T. (1959). Open-Channel Hydraulics. McGraw-Hill Book Co.

Engman, E. T. (1986). Roughness Coefficients for Routing Surface Runoff. Journal of Irrigation and Drainage Engineering, 112(1), 39–53. https://doi.org/10.1061/(asce)0733-9437(1986)112:1(39)

Feyen, H., Wunderli, H., Wydler, H., & Papritz, A. (1999). A tracer experiment to study flow paths of water in a forest soil. Journal of Hydrology, 225(3–4). https://doi.org/10.1016/S0022-1694(99)00159-6

Hinsberger, R., Biehler, A., & Yörük, A. (2022). Influence of Water Depth and Slope on Roughness—Experiments and Roughness Approach for Rain-on-Grid Modeling. Water, 14(24), 4017. https://doi.org/10.3390/w14244017

Holden, J., Kirkby, M. J., Lane, S. N., Milledge, D. G., Brookes, C. J., Holden, V., & McDonald, A. T. (2008). Overland flow velocity and roughness properties in peatlands. Water Resources Research, 44(6). https://doi.org/10.1029/2007WR006052

Masoodi, A., & Kraft, P. (2024). Investigation of Different Roughness Approaches and Vegetation Height Effects on rain-induced overland flow. Copernicus GmbH. https://doi.org/10.5194/egusphere-2024-1276

Oberle, P., Andreas, K., Tim, K., Ernesto, R. R., & Franz, N. (2021). Diskussionsbeitrag zur Fließwiderstands-parametrisierung zur Simulation von Oberflächenabflüssen infolge Starkregen. Dresdner Wasserbauliche Mitteilungen 65, Wasserbau zwischen Hochwasser und Wassermangel.

Scaini, A., Audebert, M., Hissler, C., Fenicia, F., Gourdol, L., Pfister, L., & Beven, K. J. (2017). Velocity and celerity dynamics at plot scale inferred from artificial tracing experiments and time-lapse ERT. Journal of Hydrology, 546, 28–43. https://doi.org/10.1016/j.jhydrol.2016.12.035

Tsuboyama, Y., Sidle, R. C., Noguchi, S., & Hosoda, I. (1994). Flow and solute transport through the soil matrix and

macropores of a hillslope segment. Water Resources Research, 30(4), 879–890.

https://doi.org/10.1029/93WR03245

van Verseveld, W. J., Barnard, H. R., Graham, C. B., McDonnell, J. J., Brooks, J. R., & Weiler, M. (2017). A sprinkling

experiment to quantify celerity–velocity differences at the hillslope scale. Hydrology and Earth System

Sciences, 21(11), 5891–5910. https://doi.org/10.5194/hess-21-5891-2017

---

## Author Comment (AC2)

Dear anonymous reviewer #2,

We thank the you for this positive feedback, the careful review of our manuscript and the useful comments. We will use these to improve the manuscript. We respond to each comment below (original comment in black font, our response in blue font).

This manuscript uses rainfall simulation and tracer experiments, including NaCl, Uranine, Bromide and deuterium, to understand the magnitude and spatial distribution of overland flow (OF) and topsoil return flow TIF) at two plots in a pre-Alpine catchment, one with clear-cutting, one with grassland. They also explore the celerity and velocity for OF and TIF. Overall, they put in very significant effort to conduct all the experiments. I find it is overall hard to follow all the experiments.

We thank the reviewer for recognizing the experimental effort involved in this study and for the constructive feedback on the manuscript. We would like to clarify that the "natural clearing" referred to in this study is not a result of clear-cutting, but rather a natural forest opening.

**Specific Comments:**

It seems that the mean intensity for experiments in clearing and grassland is quite different (Table 2). I would guess that might also impact the partitioning between OF and TIF, where high intensity for grassland will result in a higher OF. So, I am not sure which plays a bigger role, intensity or soil macropores.

Thank you for this pertinent comment. Indeed, a higher rainfall intensity can generate more OF, and this is probably part of the reason why we have such a difference. However, we have also monitored this location during natural rainfall events and have observed a similar distribution of OF and TIF, see table below. Thus, we do not think that the differences between the plots is mainly due to the differences in the rainfall intensity

Table 1: Runoff ratio of overland flow and topsoil interflow at the clearing and grassland locations under different natural rainfall events.

| Date of rain event | Rain depth (mm) | Mean Intensity (mm/h) | Clearing | | Grassland | |
| --- | --- | --- | --- | --- | --- | --- |
| | | | OF runoff ratio (-) | TIF runoff ratio (-) | OF runoff ratio (-) | TIF runoff ratio (-) |
| 2022-09-14 | 19.6 | 2.5 | 0.02 | 0.13 | 0.23 | 0.01 |
| 2022-09-16 | 50.6 | 2.0 | 0.45 | 0.57 | 0.81 | 0.00 |
| 2022-09-26 | 18.1 | 1.6 | 0.16 | 0.28 | 0.30 | 0.01 |
| 2022-10-01 | 33.6 | 2.0 | 0.29 | 0.40 | 0.54 | 0.01 |
| 2022-10-02 | 24.9 | 2.2 | 0.21 | 0.13 | 0.75 | 0.01 |
| 2022-10-08 | 6.3 | 0.9 | - | - | 0.17 | 0.00 |

Although we observe some variability in the runoff ratios from event to event, which are probably due to differences in antecedent moisture conditions, rain depth, intensity, the general trend supports our observation that most of the rain (simulated or not) turned into OF and not TIF.

Does that matter if two sprinklers contribute more total deuterium mass at the overlapped area (in the middle of Figure 3a)?

We thank the reviewer for this comment. While Figure 3a may have given the impression of a substantial overlap, the actual area where the sprinklers intersected was small. Additionally, the sprinklers apply less water near the edges of the application area, which means that the overlapping zone likely received only slightly more water and tracer. In theory, the local additional application of deuterium and water could generate a local pulse of water or could cause ponding on the surface. However, we did not observe this, and all water infiltrated quickly into the soil. The extra pulse of water and deuterium may also lead to some more preferential flow which would lead to a faster breakthrough than for the other areas. However, we a) would expect this to average out when looking at the full application area and b) did not use the deuterium breakthrough curve to calculate the velocity due to the uncertainty in calculating the minimum distance to the trench.

How do you determine the first increase in the flow rate in Figure 5? I think there are flow rate up and down before and after the points you labelled. Why are the locations you labelled the response to the water pulse? Thank you.

Thank you for these comments. We will explain more carefully how we selected the first responses of the water pulses in the revised document.

To determine the first increase of the flow rate, we selected the first rising limb that occurred directly after the application of the water pulses. The small increases during the long rising limb might be due to small changes in the sprinkling rate (Fig 5a) and due to variations in the pumping rate (Fig 5b, see section 4.1).

It took me a long time to really understand what Figure 6 represents: I guess you can unify with NaCl/Uranine/Bromide using red lines, and only with deuterium with grey shading in Figure 6.

For greater consistency with Figure 4, we will change the gray lines to red lines and only use the gray shading only to indicate the period of the deuterium application.

For Table 3, you can list the duration of each tracer experiment and their start time if possible.

The tracer applications were 'instantaneous' (i.e., within a minute (see Line 218), except for the deuterium applications, which took 30 minutes in the clearing and 17 minutes in the grassland (see Lines 227-229). The actual application times are indicated with red lines and gray shading in figures 4 and 6. The duration of the experiments is given in Table 2. The actual start time of the tracer applications during

the day seems fairly irrelevant and we thus prefer to not include this information in Table 3 to avoid that it looks very cluttered. The actual times are however, indicated in the datasets.

Also, if possible, add a table for sample collections and collection intervals.

We will add a table with the details about the number of samples collected and at which interval. Also, we will add information about the number of samples that were analysed.

---

## Author Response (AR1)

20th August 2025

Dear Editor,

Thank you for your comments and monitoring the publication process of our manuscript.
Please find below (in blue) the updates that we made to the manuscript in response to the reviewer's comments, as well as some minor corrections to errors we found in the previous version of the manuscript.
* * *
Comment reviewer #1

**Specific Comments**

Page 3, Line 101-104: The claim seems to be that fast flow pathways are a key factor because streamflow reacted earlier than groundwater in 'about half of the events'. This raises the question if that is special compared to other catchments, or is this typical? And how does this relate to velocity (fast flow paths) vs celerity (earlier rise)?

**We rewrote this part for greater clarity (line 101-109), following the answer below.**

A quicker rise in streamflow compared to groundwater levels is indeed observed in many catchments and is not unique to our study area (e.g., Beiter et al., 2020). Generally, this is seen as an indication that other processes than groundwater flow or subsurface flow are responsible for the initial increase in streamflow during an event, such as precipitation falling on the channel or overland flow on near stream areas.

This observation refers to celerity, i.e., the speed at which a change in pressure or water level propagates, rather than velocity, which refers to the movement of water particles or solutes, since we are discussing timing of hydrograph responses (rather than tracer movement).

Page 6, Line 185-187: Do these statistics still hold when taking the duration of rainfall into account? If the return period of 24 mm/h over one hour is 1 to 2 years, the return period of a rainfall of 24 mm/h over 24 hours will be much higher. It would be helpful if you could include this information and discuss the possible implications.

**We added additional information and the suggested comparison of the experiment for events with longer duration to the revised manuscript. Additionally, we found a mistake in the return period of the natural rain event with similar rainfall intensity than our experiments, and corrected this in the revised manuscript.**

Page 10, Line 259: Please be more specific: how were the samples selected? Were not all samples analysed for all tracers?

Only for the Bromide and Deuterium were the samples analysed in the lab. Uranine concentrations and EC were measured continuously in the field using sensors at 1-minute intervals. Not all collected samples were analysed due to financial and time constraints. During the tracer experiments, we intentionally oversampled the breakthrough curve by collecting samples at one-minute to two-minute intervals for the first hour of sampling after tracer application (see Table 1). We analysed only part of the samples to determine whether a clear tracer breakthrough could be identified. As shown in Figures 6g and S1c-d, the breakthrough curves could be very well determined by these samples already. Analysis of the additional samples collected in between the selected samples would not have provided significant new information on the breakthrough curves.

We account for any discrepancies in the time of the arrival (1 minute max if we did not analyse the absolute first sample with an elevated concentration due to sample selection) in the 2-minute uncertainty used in the velocity calculations. In Table 1, we show more clearly which samples were selected for laboratory analysis. **We added this table to the supplementary material (Table S1).**

Table 1: Overview of the samples of overland flow (OF) and topsoil interflow (TIF) that were analysed for bromide and deuterium for the tracer experiments in the clearing and the grassland. The time of application of deuterium labelled water is indicated with grey shading. The time of application of NaBr is indicated with blue shading.

| Clearing | | | Grassland | | |
|---|---|---|---|---|---|
| Date and time | Overland flow | Topsoil interflow | Date and time | Overland flow | Topsoil interflow |
| 2023-08-09 9:47 | | x | 2023-11-08 14:04 | x | x |
| 2023-08-09 9:58 | x | | 2023-11-08 14:20 | x | x |
| 2023-08-09 10:05 | | x | 2023-11-08 14:21 | x | |
| 2023-08-09 10:12 | x | | 2023-11-08 14:22 | x | x |
| 2023-08-09 12:02 | x | x | 2023-11-08 14:23 | x | |
| 2023-08-09 12:07 | x | x | 2023-11-08 14:24 | x | x |
| 2023-08-09 12:08 | | | 2023-11-08 14:25 | x | |
| 2023-08-09 12:10 | x | x | 2023-11-08 14:26 | x | x |
| 2023-08-09 12:11 | x | | 2023-11-08 14:27 | x | |
| 2023-08-09 12:12 | x | x | 2023-11-08 14:28 | x | x |
| 2023-08-09 12:13 | x | | 2023-11-08 14:29 | x | |
| 2023-08-09 12:14 | x | x | 2023-11-08 14:30 | x | x |
| 2023-08-09 12:15 | x | | 2023-11-08 14:32 | x | x |
| 2023-08-09 12:16 | x | x | 2023-11-08 14:34 | x | x |
| 2023-08-09 12:17 | x | | 2023-11-08 14:36 | x | x |
| 2023-08-09 12:18 | x | x | 2023-11-08 14:37 | | |
| 2023-08-09 12:19 | x | | 2023-11-08 14:38 | x | x |
| 2023-08-09 12:20 | x | x | 2023-11-08 14:40 | x | x |
| 2023-08-09 12:22 | x | x | 2023-11-08 14:42 | x | x |
| 2023-08-09 12:24 | x | x | 2023-11-08 14:44 | x | x |
| 2023-08-09 12:26 | x | x | 2023-11-08 14:46 | x | x |
| 2023-08-09 12:28 | x | x | 2023-11-08 14:48 | | x |
| 2023-08-09 12:30 | x | x | 2023-11-08 14:50 | x | x |
| 2023-08-09 12:32 | x | x | 2023-11-08 14:54 | x | x |
| 2023-08-09 12:34 | x | x | 2023-11-08 14:58 | x | x |
| 2023-08-09 12:36 | x | x | 2023-11-08 15:02 | x | |
| 2023-08-09 12:38 | | x | 2023-11-08 15:06 | x | x |
| 2023-08-09 12:40 | x | x | 2023-11-08 15:10 | x | x |
| 2023-08-09 12:44 | x | x | 2023-11-08 15:14 | x | x |
| 2023-08-09 12:48 | x | x | 2023-11-08 15:18 | x | x |
| 2023-08-09 12:52 | x | x | 2023-11-08 15:20 | x | |
| 2023-08-09 12:56 | x | x | 2023-11-08 15:22 | | x |
| 2023-08-09 13:00 | x | x | 2023-11-08 15:26 | | x |
| 2023-08-09 13:04 | | x | 2023-11-08 15:30 | x | x |
| 2023-08-09 13:08 | | x | 2023-11-08 15:35 | | x |

| | | | | | |
|---|---|---|---|---|---|
| 2023-08-09 13:10 | x | | 2023-11-08 15:40 | x | x |
| 2023-08-09 13:12 | | x | 2023-11-08 15:45 | | x |
| 2023-08-09 13:16 | | x | 2023-11-08 15:50 | x | x |
| 2023-08-09 13:20 | x | x | 2023-11-08 16:00 | x | x |
| 2023-08-09 13:25 | | x | 2023-11-08 16:10 | x | x |
| 2023-08-09 13:30 | x | x | 2023-11-08 16:20 | x | x |
| 2023-08-09 13:35 | | x | 2023-11-08 16:30 | x | x |
| 2023-08-09 13:40 | x | x | | | |
| 2023-08-09 13:50 | x | x | | | |
| 2023-08-09 14:00 | x | x | | | |
| 2023-08-09 14:10 | | x | | | |
| 2023-08-09 14:15 | x | | | | |
| 2023-08-09 14:20 | | x | | | |
| 2023-08-09 14:30 | x | x | | | |
| 2023-08-09 14:40 | | x | | | |
| 2023-08-09 14:45 | x | | | | |
| 2023-08-09 14:50 | | x | | | |
| 2023-08-09 15:00 | x | x | | | |
| 2023-08-09 15:10 | | x | | | |
| 2023-08-09 15:20 | | x | | | |
| 2023-08-09 15:30 | x | x | | | |
| 2023-08-09 15:40 | | x | | | |
| 2023-08-09 15:45 | | x | | | |
| 2023-08-09 16:00 | x | x | | | |
| 2023-08-09 16:15 | | x | | | |
| 2023-08-09 16:30 | x | x | | | |
| 2023-08-09 16:45 | | x | | | |
| 2023-08-09 17:00 | x | x | | | |
| 2023-08-09 17:25 | | x | | | |
| 2023-08-09 17:35 | x | | | | |
| 2023-08-09 17:55 | | x | | | |
| 2023-08-09 18:05 | x | | | | |
| 2023-08-09 18:15 | | x | | | |
| 2023-08-09 18:48 | | x | | | |
| 2023-08-09 19:00 | x | | | | |
| 2023-08-09 21:00 | x | | | | |
| 2023-08-09 23:00 | x | | | | |
| 2023-08-10 1:00 | x | | | | |
| 2023-08-10 3:00 | x | | | | |
| 2023-08-10 5:00 | x | | | | |
| 2023-08-10 7:00 | x | | | | |

| | | | | | |
|---|---|---|---|---|---|
| 2023-08-10 9:00 | x | | | | |
| 2023-08-10 10:50 | x | x | | | |
| 2023-08-10 11:50 | | x | | | |
| 2023-08-10 13:17 | | x | | | |

Page 12, Line 341: where did the remaining third of the water go?

**In the revised manuscript, we added an explanation of where the remaining third of the water likely went (line 355) following the answer below.**

The runoff ratio was 20% for OF and 46% for TIF, indicating that we collected 66% of the water applied to the plot at the trench during the experiments. The remaining one third of the water might have either percolated in deeper soil layers or flowed laterally out of the plot as they were not bound (e.g. via the "blue dye" channel in clearing leaving the plot on the right-side Fig.8.a).

Page 15, Line 398: NaBr was applied to shallow piezometers. Did you observe any overflowing of these boreholes?

**No update was made in the manuscript as we did not observe any overflow from the tubes.**

Page 17, Table 5: Why are the velocity estimates from the Deuterium results not included here?

**We did not update the manuscript regarding this comment due to the following answer below.**

We did not include Deuterium in the velocity estimates as we applied the Deuterium-labelled water diffusely via the sprinklers. The application distance to the trench ranged from a few centimeters to several meters, which makes it difficult to define a single representative travel distance to calculate the particle velocity. Although we observed Deuterium breakthrough at the trenches, the uncertainty in the travel distance makes the derived velocity estimate not directly comparable to the tracers applied in a line at a defined distance from the trench. For this reason, we chose not to include Deuterium in Table 5.

However, based on the observed first arrival times of deuterium at the trench and estimated travel distances between 0.5-10 m in the clearing and 0.5-7.5 m in the grassland, we can estimate a possible maximum velocity range of 8-150 m h$^{-1}$ for OF and 3-50 m h$^{-1}$ for TIF in the clearing and 10-150 m h$^{-1}$ for OF and 4-56 m h$^{-1}$ for TIF in the grassland.

Page 18, Line 445-454: Could you elaborate more on this topic? What does the incomplete recovery mean for your conclusions? Are the differences in tracer transport and recovery only because of the nature of the different flow paths, or are properties of the compounds also an explanation that needs to be considered. e. g, different adsorption characteristics?

**We added information to the revised manuscript (line 628-637) following the answer below.**

The tracers used in this study (NaBr, NaCl, Uranine, and Deuterium) are all commonly used in hydrological tracer experiments because they are generally considered to be conservative and exhibit minimal adsorption under typical soil conditions. In particular, the anions Br⁻ and Cl⁻, as well as Deuterium, are well known for their low reactivity and high mobility in soils (e.g., Anderson et al., 2009; Feyen et al., 1999; Scaini et al., 2017; Tsuboyama et al., 1994; van Verseveld et al., 2017). While we cannot entirely rule out small differences in transport behavior among the tracers, we do not expect these to significantly influence the recovery for our experiments. The main reasons for incomplete recovery are: 1) a portion of the tracer likely remained in the soil, 2) lateral losses (as indicated by blue dye flow paths in Figure 8), 3) percolation into deeper flow paths that we could not capture in our collection systems, and 4) measurement uncertainties (e.g., in the flow rates).

Page 19, Table 6: Unfortunately, it is not fully clear what is shown here exactly. Is the recovery expressed cumulatively? Is it the percentage of the total applied mass, or only for the first tracer lines in the first columns? Please add explanation to the table caption. Maybe consider moving the last two sentences from the caption to the discussion.

**We revised the caption of Table 6 and referred to the possible loss of tracer via the outflow in the revised manuscript in line 635.**

Page 22, Line 564: Would this not require a similar velocity of transport at both sites to be a fair comparison?

**We clarified this point in the revised manuscript (line 583-587) following the response below.**

Our statement refers to the total mass of tracer recovered in TIF over the first 100 minutes of the experiments. This comparison focuses on the overall tracer recovery integrated over the 100-minute period. Of course, very different velocities would lead to very different tracer recoveries over a certain period but in this case the focus is on recovery and therefore the comparison is still valid. In fact, the point we aimed to emphasize here is that, despite lower TIF flow rates in the grassland (and higher contributions from overland flow), the amount of tracer reaching the TIF outlet was still similar to that in the clearing. This suggests that there is concentrated tracer transport through the subsurface occurring at the grassland plot.

Page 23, Line 594 - 601: how about ambient temperature? Water viscosity changes a lot with temperature, and that influences the flow velocity.

**We acknowledged in the revised manuscript that the viscosity was different and contributed to the difference in the velocity (line 625 –629). We also added Figure S2 to the supplementary material.**

Page 25, Line 604-635: This part of the discussion could be more elaborate. The OF velocity will be determined by slope and surface characteristics (roughness, infiltrability), which in turn will be determined by the types and states of vegetation and soil. Also, the temperature (viscosity of the water) and other experimental conditions like length of the flow paths also play a role. It is thus not only 'vegetated' vs 'bare' soil. Comparing mean (see below) velocities without normalizing for these factors is not really conclusive.

**In the revised manuscript, we reformulated that part and added more information on the processes and how it could influence the velocity (line 642 – 678).**

How were the velocities averaged? Arithmetic or harmonic mean? This also applies to the other average velocities reported here. Example: When the time that overland flow needs to travel a distance of 2 m would be 1 minute and 2 minutes, the average velocity would be 1.3333 m per minute (harmonic mean).

**We decided to keep the arithmetic mean in the updated manuscript as we provide the data for the calculation of the harmonic mean.**

How would the measured flow velocities compare with theoretical estimates, e.g., Gauckler-Manning-Strickler formula? Would you get realistic roughness values when inverting the formula?

**No change was done in the manuscript as it was not possible to get realistic values from the Manning-Strickler formula (see answer reviewer I).**

Page 26, Line 685: The data should be uploaded before publication. In fact, it would be helpful if they could be included in the review. Otherwise, chances are too high that this will never happen.

**We have created the data to the WSL repository envidat.ch and provided a link to the dataset, that will be available soon (within a week), in the revised manuscript (line 738).**

**Minor Comments / Clarifications**

Page 3, Line 103: What does 'close' mean, in m?

**In the revised manuscript, we added that groundwater table depths in the catchment vary between 1.5 m and the soil surface (line 99).**

Page 5, Line 125-127: Please be a bit clearer: 10 cm organic rich AND another horizon rich in organic with 30 cm thickness, or up to 30 cm depth? Would that be A and B horizon, or litter layer and A horizon, or something else?

**We added a clear description of the horizon to the revised manuscript (line 130 – 138).**

Page 5, Line 128: Figure S1 is not about roots

**We made sure that every reference to a figure or a table in the text is correct.**

Page 8, Table 3: Tracer volumes are given for Uranine and Deuterium - what were the masses? Please specify to align with the table header

**In the revised manuscript, we changed the table header and provided clearer information regarding the amounts of applied tracers and solution volumes.**

Page 18, Figure 7: Is this the from the Deuterium experiment? Please add more info to the caption.

**We clarified this in the revised manuscript.**

Page 20, Fig 8: These are great images. Perhaps make them a bit larger (page width)?

**Thank you, we enlarged Figure 8 to the full-page size in the revised manuscript.**

Page 22, Line 543: Does this refer to Deuterium labeled water? Please clarify.

**We clarified this in the revised manuscript in line 560-562.**

Page 22, Line 562: That could possibly be exfiltration from biomat flow, right?

**We added information about the processes and highlighted that we collected both OF and biomat flow in the revised manuscript (line 580-585).**

Page 22, Line 564 – Page 23, Line 566: This requires a little more explanation. Would that mean that concentrations were much higher with the lower flow rates?

**We clarified this in the revised manuscript in line 583-587.**

Page 24, Line 609: What does this flow rate should tell the reader? Isn't the surface area/wetted perimeter equally important?

**We realize that this additional information is not important for the point that we want to make and instead of giving more information, we rewrote and shortened this sentence to avoid confusion (line 645).**

Page 25, Line 642: Please clarify why the saturated and steady-state conditions would make comparisons difficult?

**We rewrote the sentence and added arguments for greater clarity (line 684-687) following the answer below.**

In the other studies, the authors need to account for vertical flow through the unsaturated zone and especially the change in storage in the soil during soil wetting. In contrast, our study was conducted during near-saturated, steady-state conditions where storage changes were minimal.

Page 25, Line 673: info that these are 'trenched' maybe more important than the width

**We changed this sentence in the revised manuscript and mentioned trench in addition to the 8 m wide plots. We kept the 8 m wide here to indicate that the plots were relatively large.**

Page 26, Line 680: maybe include a comment on the difference in OF and TIF velocities - both are fast, but also OF still is significantly faster

**We rewrote sentences in the conclusion and added this information to the revised manuscript.**

**Technical Corrections**

**We corrected the following mistakes pointed out by reviewer #1 in the updated manuscript**

Page 6, Line 160: "(see Gauthier et al. (2025))" - Consider avoiding the double parentheses - check style guide

Page 10, Line 283: "h" - variables are set in italic, please check style guide – also variables elsewhere

Page 10, Line 270: "containing the 3 mg L-1 brilliant blue dye" – check wording/sentence structure

Page 10, Line 272: "tree" - typo

Page 10, Line 279: "was able to see" - check wording/sentence structure

Page 18, Line 450: "large" - Please check

Page 25, Line 651: Check sentence structure

**Comment reviewer # 2**

**Specific Comments:**

It seems that the mean intensity for experiments in clearing and grassland is quite different (Table 2). I would guess that might also impact the partitioning between OF and TIF, where high intensity for grassland will result in a higher OF. So, I am not sure which plays a bigger role, intensity or soil macropores.

**No change was made to the manuscript according to the answer below.**

Thank you for this pertinent comment. Indeed, a higher rainfall intensity can generate more OF, and this is probably part of the reason why we have such a difference. However, we have also monitored this location during natural rainfall events and have observed a similar distribution of OF and TIF, see table below. Thus, we do not think that the differences between the plots is mainly due to the differences in the rainfall intensity

*Table 1: Runoff ratio of overland flow and topsoil interflow at the clearing and grassland locations under different natural rainfall events.*

| Date of rain event | Rain depth (mm) | Mean Intensity (mm/h) | Clearing | | Grassland | |
|---|---|---|---|---|---|---|
| | | | OF runoff ratio (-) | TIF runoff ratio (-) | OF runoff ratio (-) | TIF runoff ratio (-) |
| 2022-09-14 | 19.6 | 2.5 | 0.02 | 0.13 | 0.23 | 0.01 |
| 2022-09-16 | 50.6 | 2.0 | 0.45 | 0.57 | 0.81 | 0.00 |
| 2022-09-26 | 18.1 | 1.6 | 0.16 | 0.28 | 0.30 | 0.01 |
| 2022-10-01 | 33.6 | 2.0 | 0.29 | 0.40 | 0.54 | 0.01 |
| 2022-10-02 | 24.9 | 2.2 | 0.21 | 0.13 | 0.75 | 0.01 |
| 2022-10-08 | 6.3 | 0.9 | - | - | 0.17 | 0.00 |

Although we observe some variability in the runoff ratios from event to event, which are probably due to differences in antecedent moisture conditions, rain depth, intensity, the general trend supports our observation that most of the rain (simulated or not) turned into OF and not TIF.

Does that matter if two sprinklers contribute more total deuterium mass at the overlapped area (in the middle of Figure 3a)?

**We modified the sprinkler area in Figure 3 in the revised manuscript as the overlapped area was small, see answer below.**

We thank the reviewer for this comment. While Figure 3a may have given the impression of a substantial overlap, the actual area where the sprinklers intersected was small. Additionally, the sprinklers apply less water near the edges of the application area, which means that the overlapping zone likely received only slightly more water and tracer. In theory, the local additional application of deuterium and water could generate a local pulse of water or could cause ponding on the surface. However, we did not observe this, and all water infiltrated quickly into the soil. The extra pulse of water and deuterium may also lead to some more preferential flow which would lead to a faster breakthrough than for the other areas. However, we a) would expect this to average out when looking at the full application area and b) did not use the deuterium breakthrough curve to calculate the velocity due to the uncertainty in calculating the minimum distance to the trench.

How do you determine the first increase in the flow rate in Figure 5? I think there are flow rate up and down before and after the points you labelled. Why are the locations you labelled the response to the water pulse? Thank you.

**Thank you for these comments. We explained more carefully how we selected the first responses of the water pulses in the revised manuscript.**

It took me a long time to really understand what Figure 6 represents: I guess you can unify with NaCl/Uranine/Bromide using red lines, and only with deuterium with grey shading in Figure 6.

**For greater consistency with Figure 4, we changed the grey lines to red lines and only used the grey shading to indicate the period of the deuterium application.**

For Table 3, you can list the duration of each tracer experiment and their start time if possible.

**No change was made to the manuscript according to the below answer.**

The tracer applications were 'instantaneous' (i.e., within a minute (see Line 218), except for the deuterium applications, which took 30 minutes in the clearing and 17 minutes in the grassland **(see Lines 238-240)**. The actual application times are indicated with red lines and gray shading in figures 4 and 6. The duration of the experiments is given in Table 2. The actual start time of the tracer applications during the day seems fairly irrelevant and we thus prefer to not include this information in Table 3 to avoid that it looks very cluttered. The actual times are however, indicated in the datasets.

Also, if possible, add a table for sample collections and collection intervals.

**We added a table with the details about the samples which were analysed for bromide and deuterium to the supplementary material (Table S1).**

**Updates from personal mistakes**

We have updated some figures, tables, and typographical errors that we found in the text to improve the clarity and the correctness of the manuscript.
We have corrected the return period of the natural events with similar rainfall intensity to the experiment, after noticing an error thanks to the comment of reviewer #1. We also corrected the percentage of the deuterium-enriched water that we used for the tracer experiments.
These revisions result in minor changes and do not affect the significance or interpretation of the results.

---

## Author Response (AR2)

Dear reviewer, dear Jan Wienhöfer,

We thank you for your valuable feedback, the careful review of our manuscript, and the useful comments. We have used these to improve the manuscript. We respond to each comment below (original comment in black font, our response in blue font).

The authors have adequately addressed almost all of the reviewers' comments, and revised their manuscript accordingly. I suggest publication of the manuscript after addressing the remaining points below.

Thank you for this positive assessment.

Page 7, line 194 (maximum of 50 mm/h) – what does that addition in parentheses mean regarding the 10-years rain, which is about 39 mm/h?

Thank you for this comment. The parenthesis '(maximum of 50 mm/h)' meant that the maximum rainfall intensity per hour based on 38 years of hourly precipitation data from the Erlenhöhe meteorological station was 50 mm/h. **We now write this explicitly '(maximum rainfall intensity recorded: 50 mm/h)' in the revised manuscript.**

Page 19, Table 6: Unfortunately, it is still not fully clear what is shown here exactly. If the recovery is expressed cumulatively, as the caption suggests, why does the recovery for Uranine in OF at the clearing drop from 26 % to 17 %?

Thank you for this comment. The recovery (in % of applied mass) of, for example, Uranine in OF in the clearing is less after 24 hours compared to 163 minutes because the total applied tracer mass increased after 163 minutes when the second tracer line was added to the plot.

**We have revised the caption of Table 6 accordingly:**

Table 6: Cumulative tracer recovery as percentage of the applied mass for each tracer used in the experiments for the plot in the clearing and the plot in the grassland. For the plot in the clearing, the second lines of NaCl and uranine were applied 163 minutes after the first applications. Therefore, the values reported for 100 and 163 minutes include only the recovery of the first tracer application, while the values reported for 24 hours include the recovery from both applications. Because the total applied mass increased with the second application and the recovery for the second line (applied further upslope) was less than for the first line, the cumulative recovery after 24 hours expressed as a percentage of the total applied mass is less than after 163 minutes. Some of the tracer applied to the upper parts of the plot in the clearing likely left via an outflow on the side of the plot (see section 4.4). This affected the recovery of NaCl 2, uranine 2, NaBr, and deuterium-labelled water that were applied upslope from this outflow. BDL stands for "below detection limit".

Page 27, Line 737: The data cannot be accessed using the doi "10.16904/envidat.685", only a loading screen is shown (tested on 26 Sep 2025). The doi resolves to https://envidat.ch/#/metadata/flow-and-tracer-time-series-for-overland-flow-and-topsoil-interflow-during-rainf [!]. Is perhaps something missing here? Please make sure that the data can be accessed.

Thank you for this comment. Indeed, it may appear that the letters "all" are missing from "rainfall" but the url is correct (and automatically generated based on a maximum number of characters). Also, the website was under maintenance with the following message: 'Maintenance Mode
We are currently upgrading EnviDat backend. Thank you for your understanding and patience during this time. EnviDat can be accessed in read-only mode. Data download, upload and user data management functionalities will be disabled.' (30.09.2025)

This is why you might not have been able to access the data previously. **We now retested the link and everything seems to be correct and working. (08.10.2025)**